# BPDQ: Bit-Plane Decomposition Quantization on a Variable Grid for Large Language Models

**Junyu Chen** [1 2 3]  **Jungang Li** [4]  **Jing Xiong** [2]  **Wenjie Wang** [1 3]  **Qingyao Yang** [2]  **He Xiao** [2]  **Zhen Li** [5 6]
**Taiqiang Wu** [2]  **Mengzhao Chen** [2]  **Zhen Peng** [7]  **Chaofan Tao** [2]  **Long Shi** [1 3]  **Hongxia Yang** [5 6]  **Ngai Wong** [2]

## Abstract

Large language model inference is often bounded by memory footprint and bandwidth in resource-constrained deployments, making quantization fundamental to efficient serving. While post-training quantization (PTQ) maintains high fidelity at 4-bit, it deteriorates at 2-3 bits. In essence, existing methods enforce a shape-invariant quantization grid (e.g., the fixed uniform intervals of UINT2) for each group, severely restricting the feasible set for error minimization. To address this, we propose Bit-Plane Decomposition Quantization (BPDQ), which constructs a variable quantization grid via bit-planes and scalar coefficients, and iteratively refines them using second-order information while progressively compensating for quantization errors to minimize output discrepancy. In the 2-bit regime, BPDQ enables serving Qwen2.5-72B on a single RTX 3090 with 83.85% GSM8K accuracy (vs. 90.83% at 16-bit). Moreover, we theoretically show that the variable grid expands the feasible set, and that the quantization process consistently aligns with the optimization objective in Hessian-induced geometry. The code is available at GitHub.

## 1. Introduction

Large language models (LLMs) demand substantial memory and compute resources, making efficiency a major research focus in academia and industry (Miao et al., 2023; Zhu et al., 2024). Among efficiency approaches, quantization is a fundamental technique that reduces memory footprint and alleviates memory bandwidth bottlenecks during inference (Gong et al., 2024; Zhou et al., 2024). Accordingly, many recent open-source models release low-bit checkpoints. For example, Qwen3 offers an official 4-bit quantized variant (Qwen Team, 2025), suggesting that 4-bit weight-only quantization preserves high fidelity. Specifically, quantization-aware training (QAT) demonstrates promising performance by learning directly in the low-bit space, yet incurs prohibitive training cost (Liu et al., 2023; Chen et al., 2024). Furthermore, quantization-aware fine-tuning (QAF) can improve low-bit performance by fine-tuning a quantized model, but it requires a two-stage pipeline (Dettmers et al., 2023; Xu et al., 2023; Chen et al., 2025a). For post-training quantization (PTQ), distribution-aware methods utilize weight or activation statistics to reduce distortion induced by outliers (Lin et al., 2024; Ashkboos et al., 2024), which rely on handling outliers during inference. In contrast, optimization-based methods such as GPTQ (Frantar et al., 2022; Zhang et al., 2025a) are theoretically well grounded by minimizing output discrepancy under an output-aligned objective (e.g., $\|\mathbf{WX} - \widehat{\mathbf{W}}\mathbf{X}\|$) (Zhang et al., 2025a; Chen et al., 2025b), while preserving hardware-friendly inference.

Nevertheless, pushing precision down to 2-3 bits remains challenging because of limited cardinality (e.g., 2-bit offers only four distinct values), which causes significant representational loss and severe degradation in model quality. In the exploration of low-bit quantization, distribution-aware methods (Huang et al., 2024a;b; Li et al., 2024) apply hybrid formats or mixed precision to protect salient weights, leading to irregular memory access patterns. Vector Quantization (VQ) methods (Liu et al., 2024; Egiazarian et al., 2024) achieve high fidelity by mapping weights to codebooks, but suffer from prohibitive computational costs during codebook optimization. While bit-plane methods (Park et al., 2025; Tran & Nguyen, 2025) enable accelerator-friendly bit-parallel arithmetic, they lack a rigorous output-aligned objective and rely on fine-tuning to preserve fidelity.

Despite these explorations, optimization-based PTQ main-

---

[1]Southwestern University of Finance and Economics [2]The University of Hong Kong [3]Artificial Intelligence and Digital Finance Key Laboratory of Sichuan Province [4]The Hong Kong University of Science and Technology (Guangzhou) [5]The Hong Kong Polytechnic University [6]InfiX.ai [7]Sun Yat-sen University. Correspondence to: Junyu Chen <223081200039@smail.swufe.edu.cn>, Long Shi <shilong@swufe.edu.cn>, Hongxia Yang <hongxia.yang@polyu.edu.hk>, Ngai Wong <nwong@eee.hku.hk>.

*Proceedings of the 43$^{rd}$ International Conference on Machine Learning*, Seoul, South Korea. PMLR 306, 2026. Copyright 2026 by the author(s).

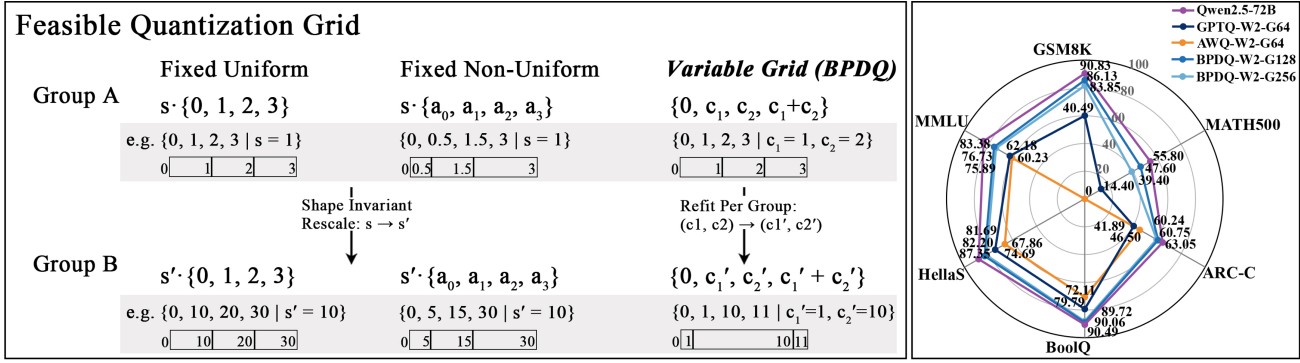

*Figure 1.* (a) Fixed grids (Uniform/Non-Uniform) enforce shape invariance, where the relative spacing of quantization levels is shared across groups (scaled by $s$). BPDQ breaks this limitation by constructing a variable grid per group using bit-plane coefficients ($c_1$, $c_2$), expanding the feasible set. (b) Performance comparison of 2-bit quantized Qwen2.5-72B.

tains a rigorous theoretical formulation but fails in the low-bit regime. We attribute this failure to a critical misalignment between the optimization objective and the rigid quantization grid. ***Essentially, the problem is not a failure of the optimization objective, but the rigidity of the quantizer's feasible set under that objective.*** As illustrated in Figure 1 (a), a fixed uniform grid restricts the per-group feasible values to scale $\cdot \{0, 1, 2, 3\}$, while a fixed non-uniform grid employs scale $\cdot \{a_0, a_1, a_2, a_3\}$. Although the scale varies across groups, the relative spacing pattern of the four levels is shared across all groups, making each group only a magnified or shrunken copy of the same template. This shape invariance can be overly restrictive, since the output-aligned objective is the nearest-point projection in the Hessian-induced geometry.

To address this limitation, we propose Bit-Plane Decomposition Quantization (BPDQ), which constructs a variable grid via bit-planes and scalar coefficients. The right side of Figure 1 (a) shows that BPDQ allows the relative spacing pattern to vary across groups using distinct coefficients. This breaks the shape invariance constraint and enlarges the feasible set under the output-aligned objective. Specifically, BPDQ initializes the variable grid through bit-plane decomposition and a closed-form solution for scalar coefficients. Within the Hessian-induced geometry, we iteratively refine the bit-planes and scalar coefficients. Moreover, to maintain error-propagation consistency, we introduce a delta correction, ensuring iterations align with the optimization objective. Appendix A formalizes how this variable grid expands the feasible solution set, and Appendix B formalizes the consistency of BPDQ with Hessian-induced optimality.

We validate BPDQ on the Qwen-3/2.5 family (0.6B-72B) and Ministral-3 (3B, 8B) across five language modeling and commonsense benchmarks. Furthermore, we demonstrate BPDQ's robustness on quantization-sensitive reasoning tasks and long-context benchmarks. Figure 1 (b) compares 2-bit quantization on Qwen2.5-72B, where GPTQ and

AWQ suffer severe degradation (e.g., dropping below 41% on GSM8K) while BPDQ preserves the high fidelity of the full-precision baseline (e.g., 86.13% on GSM8K). In terms of deployment efficiency, BPDQ enables serving the quantized 72B model (W2-G256) on a single RTX 3090 (22.69 GB VRAM) with 83.85% accuracy on GSM8K. By implementing a bit-plane look-up table (LUT) kernel (Park et al., 2022), we achieve low-latency decoding for real-time interactive generation. Analysis of activation statistics confirms that BPDQ inherently preserves essential outliers, which is crucial for maintaining model quality. Our contributions are summarized as follows:

- **Insight**: We identify the shape invariance of fixed quantization grids as the fundamental constraint on optimization-based PTQ in low-bit regimes. To address this, we propose BPDQ, which decomposes weights into bit-planes to construct a variable quantization grid, theoretically expanding the feasible solution set for output-aligned error minimization.

- **Methodology**: We formulate a rigorous optimization framework that extends optimization-based PTQ to variable grids. By iteratively refining bit-planes and scalar coefficients within the Hessian-induced geometry, BPDQ ensures that the optimization process consistently aligns with the optimization objective, as formally established in our theoretical analysis.

- **Performance**: Extensive experiments across language understanding, reasoning, and long-context tasks demonstrate BPDQ's consistently high fidelity in low-bit regimes. Furthermore, we provide system efficiency profiling to validate the hardware efficiency of bit-plane methods, and confirm that BPDQ inherently preserves critical outliers through activation analysis.

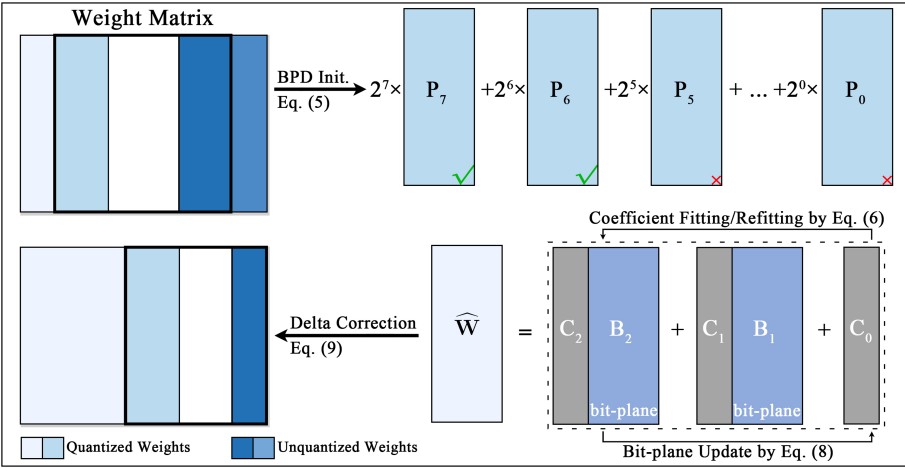

*Figure 2.* Overview of the 2-bit BPDQ quantization procedure.

## 2. Related Work

**Low-bit Quantization for LLMs.** To achieve extreme compression rates, QAT methods optimize in the Boolean domain or utilize factorized representations (Tran & Nguyen, 2025; Lee et al., 2025), albeit at substantial training costs. Among PTQ methods, vector quantization (VQ) maps weights to codebooks (Egiazarian et al., 2024; Liu et al., 2024), preserving high fidelity but suffering from prohibitive quantization overheads. Alternatively, distribution-aware methods employ hybrid formats or mixed precision to protect salient weights (Huang et al., 2024a;b; Li et al., 2024), often causing irregular memory access patterns. Recently, bit-plane and ternary decomposition methods have emerged to enable accelerator-friendly arithmetic (Xiao et al., 2025; Yan et al., 2025; Park et al., 2025). However, these approaches typically rely on progressive residual correction or fine-tuning, lacking a rigorous output-aligned objective.

**Optimization-based PTQ for LLMs.** Optimization-based PTQ minimizes output discrepancy under objectives such as $\|\mathbf{W}\mathbf{X} - \widehat{\mathbf{W}}\mathbf{X}\|$ (Zhang et al., 2025a; Chen et al., 2025b). This perspective traces back to second-order sensitivity analyses (OBD/OBS) (LeCun et al., 1989; Hassibi et al., 1993) and is further developed by Optimal Brain Compression (OBC) (Frantar & Alistarh, 2022). GPTQ employs efficient approximate second-order information for LLM quantization (Frantar et al., 2022). Recent theoretical advances connect GPTQ to Babai's nearest-plane algorithm on a Hessian-induced lattice, offering a geometric interpretation of its error propagation (Chen et al., 2025b). Furthermore, theoretical analyses have established provable error bounds for these procedures (Zhang et al., 2025a), while enhanced sequential solvers like Qronos (Zhang et al., 2025b) have been introduced to integrate past-error correction to further minimize reconstruction loss. However, the rigidity of fixed quantization grids restricts the feasible

solution set for optimization-based PTQ, leading to degradation in the low-bit regime. To overcome this restriction, BPDQ constructs a variable grid that expands the feasible set, achieving high fidelity at extreme compression rates.

## 3. Methodology

BPDQ follows the optimization objective while replacing the fixed quantization grid with a variable grid. Within the Hessian-induced geometry, BPDQ initializes via bit-plane decomposition (BPD) and closed-form scalar coefficient fitting. It then iteratively refines the grid through column-wise bit-plane update and group-wise scalar coefficient refitting with Hessian-aware error compensation. Finally, it applies a delta correction to maintain error-propagation consistency. The formal consistency of this procedure with Hessian-induced optimality is established in Appendix B.

Specifically, the variable grid quantized weight $\widehat{\mathbf{W}}$ is formed by bit-planes and scalar coefficients:

$$\widehat{\mathbf{W}} = \text{REP}(\mathbf{C}_0) + \sum_{i=1}^{k} \text{REP}(\mathbf{C}_i) \odot \mathbf{B}_i, \qquad (1)$$

where $\mathbf{B}_i \in \{0,1\}^{d_{\text{out}} \times d_{\text{in}}}$ (for $i = 1, \ldots, k$) is the $i$-th bit-plane, $\mathbf{C}_i \in \mathbb{R}^{d_{\text{out}} \times (d_{\text{in}}/g)}$ is the group-wise scale coefficient with group size $g$, and $\mathbf{C}_0 \in \mathbb{R}^{d_{\text{out}} \times (d_{\text{in}}/g)}$ is the group-wise bias coefficient. The operator $\text{REP}(\cdot)$ expands $\mathbf{C}_i$ along the input dimension by repeating each group coefficient across its $g$ columns. The integer $k$ is the number of non-bias bit-planes, and $\odot$ is element-wise multiplication.

### 3.1. Preliminaries

**Optimization Objective.** Consider a linear layer with weight $\mathbf{W} \in \mathbb{R}^{d_{\text{out}} \times d_{\text{in}}}$ and input activations $\mathbf{X} \in \mathbb{R}^{d_{\text{in}} \times N}$

computed from $N$ calibration samples. The quantized weight $\widehat{\mathbf{W}} \in \mathcal{Q}$ is obtained by minimizing the output reconstruction error:

$$\begin{aligned} \widehat{\mathbf{W}} &= \underset{\widetilde{\mathbf{W}} \in \mathcal{Q}}{\arg\min} \left\| (\mathbf{W} - \widetilde{\mathbf{W}})\mathbf{X} \right\|_F^2 \\ &= \underset{\widetilde{\mathbf{W}} \in \mathcal{Q}}{\arg\min} \operatorname{tr}\!\left( (\mathbf{W} - \widetilde{\mathbf{W}})\, \mathbf{H}\, (\mathbf{W} - \widetilde{\mathbf{W}})^\top \right), \end{aligned} \quad (2)$$

where $\mathbf{H} = \mathbf{X}\mathbf{X}^\top$ is an approximate Hessian metric induced by calibration data, and $\mathcal{Q}$ denotes the set of admissible low-bit weight matrices.

**Quantization Error Compensation.** Due to the enormous parameter space of LLMs, repeatedly updating the inverse Hessian is prohibitively expensive. To address this, GPTQ (Frantar et al., 2022) operates within the Hessian-induced geometry by using the upper-triangular Cholesky factorization $\mathbf{U} = \operatorname{chol}(\mathbf{H}^{-1})$ (i.e., $\mathbf{H}^{-1} = \mathbf{U}^\top \mathbf{U}$), and performs error propagation via triangular updates to compensate the induced quantization error on the remaining free coordinates. When quantizing the $l$-th column, let $\mathbf{W}_{:,l}$ and $\widehat{\mathbf{W}}_{:,l}$ denote the current working and quantized column vectors, respectively. Then define the error coordinate:

$$\mathbf{E}_{:,l} = \frac{\mathbf{W}_{:,l} - \widehat{\mathbf{W}}_{:,l}}{\mathbf{U}_{l,l}}. \quad (3)$$

The Hessian-aware compensation update is:

$$\mathbf{W}_{:,\,l:} \leftarrow \mathbf{W}_{:,\,l:} - \mathbf{E}_{:,l}\, \mathbf{U}_{l,l:}, \quad (4)$$

which maintains the optimization-based PTQ error-propagation state under the Hessian-induced geometry.

### 3.2. Variable Grid Initialization

**Bit-Plane Selection.** Consider a contiguous column group $\mathbf{W}_{:,s:(s+g)} \in \mathbb{R}^{d_{\text{out}} \times g}$, where $s$ is the starting column index and $g$ is the group size. Applying a per-group affine quantizer with round-to-nearest (RTN) to $\mathbf{W}_{:,\,s:(s+g)}$ obtains an unsigned 8-bit integer matrix $\mathbf{Z} \in \{0, \ldots, 255\}^{d_{\text{out}} \times g}$. Then $\mathbf{Z}$ admits the bit-plane decomposition (BPD):

$$\mathbf{Z} = \sum_{i=0}^{7} 2^i\, \mathbf{P}_i, \quad (5)$$

where $\mathbf{P}_i \in \{0,1\}^{d_{\text{out}} \times g}$ is $i$-th bit-plane of $\mathbf{Z}$. For bit-plane initialization, select the $k$ most significant bit (MSB) planes. The bit-planes $(\mathbf{B}_i)_{:,s:(s+g)} = \mathbf{P}_{7-k+i}$ for $i \in \{1, \ldots, k\}$, since the MSB planes capture the dominant magnitude information. The remaining least significant bit (LSB) planes $\{\mathbf{P}_{7-k}, \ldots, \mathbf{P}_0\}$ are discarded. Removing

them introduces only a small truncation error, providing a low-error initialization.

**Scalar Coefficient Fitting.** When the bit-planes $\{(\mathbf{B}_i)_{:,s:(s+g)}\}_{i=1}^k$ are fixed, $\widehat{\mathbf{W}}$ is linear in the scalar coefficients, enabling a closed-form fit under the Hessian-induced geometry to align with the optimization objective in Eq. (2). Concretely, consider a column group $\mathbf{W}_{:,s:(s+g)}$, and let $\mathbf{U}_{\text{loc}} = \mathbf{U}_{s:(s+g),s:(s+g)} \in \mathbb{R}^{g \times g}$ be the local triangular factor of this group. For $r$-th row, define $\mathbf{B}_r = \left[ \mathbf{1}, (\mathbf{B}_1)_{r,s:(s+g)}^\top, \ldots, (\mathbf{B}_k)_{r,s:(s+g)}^\top \right] \in \{0,1\}^{g \times (k+1)}$, where $\mathbf{1}$ is the all-ones column vector. The group-wise coefficient vector $c_r \in \mathbb{R}^{k+1}$ is obtained by the following row-wise weighted least-squares fit:

$$c_r = \underset{c \in \mathbb{R}^{k+1}}{\arg\min} \left\| \mathbf{U}_{\text{loc}}^{-\top} \left( \mathbf{B}_r\, c - \mathbf{W}_{r,s:(s+g)}^\top \right) \right\|_2^2. \quad (6)$$

This scalar coefficient fitting is an optimal projection under the Hessian-induced geometry for the fixed bit-planes, yielding coefficients consistent with the output reconstruction objective and inducing the variable grid by $c_r$. In implementation, a damping factor $\alpha = 10^{-4}$ is applied for numerical stability (omitted in Eq. 6 for brevity).

### 3.3. Iteration under the Optimization Objective

For each group, BPDQ alternates bit-plane update and coefficient refitting, with delta correction for propagation-state consistency. In our experiments, we consistently set the number of iterations to 10, retaining the iterate that minimizes the group-wise propagation error $\|\mathbf{E}_{:,\,s:(s+g)}\|_F^2$.

**Bit-plane Update.** Given the fixed group-wise scalar coefficients $\{(\mathbf{C}_i)_{:,s/g}\}_{i=0}^k$, the bit-planes $\{(\mathbf{B}_i)\}_{i=1}^k$ are updated column by column via exact enumeration under the Hessian-induced geometry. For a column $l \in \{s, \ldots, s+g-1\}$ and a row $r$, enumerating bit vectors $\mathbf{b} = (b_1, \ldots, b_k) \in \{0,1\}^k$ generates $2^k$ candidate values:

$$v_r(\mathbf{b}) = (\mathbf{C}_0)_{r,s/g} + \sum_{i=1}^k (\mathbf{C}_i)_{r,s/g}\, b_i. \quad (7)$$

The optimal bit vector $\mathbf{b}^*$ is selected to minimize the local reconstruction error:

$$\mathbf{b}^* = \underset{\mathbf{b} \in \{0,1\}^k}{\arg\min} \left( \mathbf{W}_{r,l}' - v_r(\mathbf{b}) \right)^2, \quad (8)$$

where $\mathbf{W}_{:,l}'$ denotes the current working column after previous propagation updates. This minimization is executed in parallel for all rows to determine the quantized column vector $\widehat{\mathbf{W}}_{:,l}$. The update is performed column-wise and

followed by an error propagation step at each column, consistent with the formulation in Eq. (4).

**Coefficient Refitting.** After completing the bit-plane update for the entire group $\{s, \ldots, s + g - 1\}$, the group-wise scalar coefficients $\{(\mathbf{C}_i)_{:,s/g}\}_{i=0}^{k}$ are refit to match the original weights by solving the row-wise weighted least-squares problem in Eq. (6) with the updated bit-planes $\{(\mathbf{B}_i)_{:,s:(s+g)}\}_{i=1}^{k}$ fixed. This closed-form refit updates the variable grid while remaining aligned with the Hessian-induced objective in Eq. (2).

**Delta Correction.** After refitting the coefficients, we apply a delta correction to keep the error-propagation state consistent. Refitting the coefficients changes the quantized weight block from $\widehat{\mathbf{W}}_{\text{old}} \in \mathbb{R}^{d_{\text{out}} \times g}$ to $\widehat{\mathbf{W}}_{\text{new}} \in \mathbb{R}^{d_{\text{out}} \times g}$ for the current group. Here, $\widehat{\mathbf{W}}_{\text{old}}$ is obtained from the bit-plane update step, and $\widehat{\mathbf{W}}_{\text{new}}$ is the updated block after refitting the scalar coefficients. This discrepancy renders the accumulated propagation state inconsistent. Thus, we compute a correction $\Delta \mathbf{E}$ using the local triangular factor $\mathbf{U}_{\text{loc}}$:

$$\Delta \mathbf{E} \, \mathbf{U}_{\text{loc}} = \widehat{\mathbf{W}}_{\text{old}} - \widehat{\mathbf{W}}_{\text{new}}, \qquad (9)$$

where $\Delta \mathbf{E} \in \mathbb{R}^{d_{\text{out}} \times g}$ is the group-wise correction in the propagation coordinates. The coordinates are updated as $\mathbf{E}'_{:,s:(s+g)} = \mathbf{E}_{:,s:(s+g)} + \Delta \mathbf{E}$, where $\mathbf{E}_{:,s:(s+g)}$ represents the propagation error vectors computed during the bit-plane update phase. This delta correction maintains error-propagation consistency within the Hessian-induced geometry, ensuring that all iterates adhere to the same output-aligned optimization objective. A formal equivalence proof is given in Appendix B.3.

# 4. Experiments

## 4.1. Experimental Setup

**Models and Tasks.** Experiments are conducted on several large language models, including the Qwen-3 family (0.6B, 4B, 8B, 14B, 32B) (Yang et al., 2025), Qwen-2.5 (7B, 72B) (Qwen Team, 2024), and Ministral-3 (3B, 8B) (Mistral AI, 2025). Quantization quality is assessed using `lm-evaluation-harness` (Gao et al., 2024) across the following benchmarks: WikiText-2 (Merity et al., 2016), GSM8K (5-shot) (Cobbe et al., 2021), MATH500 (4-shot) (Lightman et al., 2023), ARC-C (Clark et al., 2018), BoolQ (Clark et al., 2019), HellaSwag (Zellers et al., 2019), MMLU (Hendrycks et al., 2020), and LongBench (Bai et al., 2024).

**Baselines and Hyperparameters.** BPDQ is implemented within the `GPTQModel` library (ModelCloud.ai, 2024), which also supports GPTQ (Frantar et al., 2022) and AWQ

(Lin et al., 2024). All methods employ asymmetric quantization, calibrated on 1024 samples from the C4 dataset (Raffel et al., 2019). To ensure fair comparisons at similar bits-per-weight (BPW), larger group sizes are used in BPDQ to offset the storage overhead introduced by per-bit-plane scalar coefficients. Specifically, GPTQ and AWQ use a group size of $g = 64$ for 4-bit and $g \in \{32, 64\}$ for 2/3-bit, whereas BPDQ uses $g = 128$ for 4-bit and $g \in \{64, 128\}$ for 2/3-bit. Regarding error propagation, GPTQ utilizes `desc_act` to sort channels in descending order of approximate Hessian values. Meanwhile, BPDQ employs Group-Aware Reordering (GAR) (Gafni et al., 2025) to preserve group integrity for scalar derivation, with the damping factor $\alpha$ set to $10^{-4}$ and iterations set to 10 across all experiments. Additionally, the recent bit-plane method AnyBCQ (Park et al., 2025) and the vector quantization (VQ) method VPTQ (Liu et al., 2024) are included. AnyBCQ follows its paper-recommended settings with fixed-precision configurations at 2-4 bits, while VPTQ is evaluated using officially released checkpoints.

## 4.2. Main Results

**Benefits of Variable Grid.** Table 1 shows the results across three model sizes (8B, 32B, 72B) and five quantization settings on seven benchmarks. BPDQ yields the best performance in most cases, exhibiting a significant lead over GPTQ and AWQ, particularly in the 2-bit regime. Specifically, on reasoning tasks such as GSM8K and MATH500, 2-bit AWQ suffers catastrophic collapse (e.g., 0.00% for the 2-bit 72B model), and 2-bit GPTQ shows severe deterioration. Conversely, BPDQ preserves reasoning capabilities, achieving 87.72% on GSM8K (Qwen2.5-72B W2-G64) and far surpassing GPTQ's 63.46%. Notably, although AWQ performs competitively at 3-4 bits by focusing on outlier preservation, its failure at 2-bit suggests that outlier protection alone is insufficient when the quantization grid is extremely coarse. Meanwhile, GPTQ, which shares the same Hessian-based optimization framework as BPDQ, outperforms AWQ at 2-bit but remains constrained by the fixed uniform grid. By relaxing this restriction with a variable grid, BPDQ attains superior performance by expanding the feasible solution set, which allows the Hessian-based solver to align more closely with the optimization objective. ***This comparison validates our insight: the primary restriction at ultra-low bitwidths is not a failure of the optimization objective, but the rigidity of the fixed grid.***

In the extreme compression scenario of W2-G256, BPDQ compresses Qwen2.5-72B to 22.69 GB, unlocking deployment on a single RTX 3090. Concurrently, it achieves 83.85% on GSM8K, retaining 92.32% of the baseline accuracy. Moreover, it maintains high fidelity across diverse domains, preserving over 91.01% of the baseline performance on general benchmarks (BoolQ, ARC-C, HellaSwag, MMLU), with BoolQ peaking at 99.15%.

*Table 1.* Evaluation results of Ministral3-8B, Qwen3-32B, and Qwen2.5-72B across seven benchmarks. Best and second-best results are highlighted in **bold** and underlined, respectively. Additional results for other model sizes are provided in Appendix C.

| Model | BPW | Wiki2 ↓ | GSM8K ↑ | MATH500 ↑ | ARC-C ↑ | BoolQ ↑ | HellaS ↑ | MMLU ↑ |
|---|---|---|---|---|---|---|---|---|
| *Ministral3-8B* | *16* | *9.72* | *85.90%* | *54.00%* | *64.08%* | *85.78%* | *78.80%* | *73.02%* |
| GPTQ-W4-G64 | 4.31 | **9.94** | 84.84% | 51.20% | 63.82% | 85.84% | 78.37% | 72.71% |
| AWQ-W4-G64 | 4.31 | 9.97 | 83.40% | **52.40%** | 62.97% | 85.50% | 78.24% | 72.50% |
| BPDQ-W4-G128 | 4.63 | 9.95 | 84.99% | 51.80% | 63.74% | 85.90% | **78.40%** | 72.91% |
| GPTQ-W3-G32 | 3.59 | 10.56 | 79.83% | 43.60% | 59.47% | 84.86% | **77.66%** | 70.48% |
| AWQ-W3-G32 | 3.59 | 10.75 | **81.50%** | 48.60% | **61.35%** | 85.47% | 76.18% | 69.89% |
| BPDQ-W3-G64 | 4.00 | **10.49** | 80.06% | 45.40% | 60.49% | 86.88% | 76.91% | 70.23% |
| GPTQ-W3-G64 | 3.30 | 10.85 | 76.80% | 38.80% | 59.73% | 84.80% | 77.28% | **70.07%** |
| AWQ-W3-G64 | 3.30 | 11.03 | 77.41% | **46.60%** | 60.32% | 85.75% | 75.88% | 69.26% |
| BPDQ-W3-G128 | 3.50 | **10.68** | 79.15% | 43.60% | 60.84% | 85.84% | 76.78% | 69.28% |
| GPTQ-W2-G32 | 2.56 | 19.20 | 12.36% | 2.40% | 41.47% | 62.51% | 66.66% | 45.39% |
| AWQ-W2-G32 | 2.56 | 5.3E+5 | 0.00% | 0.00% | 35.84% | 52.14% | 42.80% | 26.16% |
| BPDQ-W2-G64 | 2.75 | **14.69** | 42.46% | 17.40% | 51.62% | 84.04% | 68.06% | 60.13% |
| GPTQ-W2-G64 | 2.28 | 26.15 | 1.52% | 2.80% | 35.58% | 53.70% | 60.20% | 31.90% |
| AWQ-W2-G64 | 2.28 | 1.5E+6 | 0.00% | 0.00% | 26.45% | 38.07% | 28.50% | 26.61% |
| BPDQ-W2-G128 | 2.38 | **15.64** | 38.74% | 12.40% | 50.09% | 83.52% | 67.15% | 58.59% |
| *Qwen3-32B* | *16* | *9.34* | *74.15%* | *54.00%* | *61.01%* | *86.39%* | *82.56%* | *80.69%* |
| GPTQ-W4-G64 | 4.31 | 9.53 | 72.18% | 50.40% | 60.84% | 87.83% | **82.36%** | 79.92% |
| AWQ-W4-G64 | 4.31 | 10.23 | 74.47% | 53.00% | 60.35% | **88.35%** | 81.96% | 78.46% |
| BPDQ-W4-G128 | 4.63 | **9.52** | 76.95% | 53.60% | 62.54% | 87.55% | 82.31% | **80.07%** |
| GPTQ-W3-G32 | 3.59 | 9.95 | 56.86% | 49.60% | 57.25% | 87.77% | 81.34% | 78.26% |
| AWQ-W3-G32 | 3.59 | 10.11 | 80.14% | 50.60% | 59.90% | 84.13% | 80.93% | 78.70% |
| BPDQ-W3-G64 | 4.00 | **9.86** | 86.13% | 52.60% | 61.18% | 88.01% | 81.56% | 78.83% |
| GPTQ-W3-G64 | 3.30 | 10.14 | 46.40% | 47.40% | 56.57% | 84.62% | 81.14% | 77.06% |
| AWQ-W3-G64 | 3.30 | 10.34 | 66.19% | 51.60% | 57.94% | 86.24% | 80.65% | 76.95% |
| BPDQ-W3-G128 | 3.50 | **9.97** | 67.85% | 53.20% | 60.41% | 88.17% | 81.16% | 77.89% |
| GPTQ-W2-G32 | 2.56 | 14.64 | 44.20% | 15.80% | 39.93% | 74.16% | 69.95% | 50.08% |
| AWQ-W2-G32 | 2.56 | 8.2E+2 | 0.00% | 0.00% | 27.13% | 80.64% | 67.20% | 58.14% |
| BPDQ-W2-G64 | 2.75 | **12.34** | 80.89% | 42.40% | 56.83% | 86.85% | 76.67% | 73.24% |
| GPTQ-W2-G64 | 2.28 | 18.26 | 5.91% | 3.80% | 32.68% | 60.46% | 63.84% | 36.77% |
| AWQ-W2-G64 | 2.28 | 3.3E+7 | 3.18% | 7.20% | 31.40% | 60.09% | 47.96% | 50.14% |
| BPDQ-W2-G128 | 2.38 | **12.97** | 70.43% | 33.60% | 52.56% | 87.13% | 75.10% | 71.31% |
| *Qwen2.5-72B* | *16* | *4.72* | *90.83%* | *55.80%* | *63.05%* | *90.49%* | *87.35%* | *83.38%* |
| GPTQ-W4-G64 | 4.31 | 5.01 | 90.52% | 56.00% | **63.99%** | **90.76%** | 87.04% | 82.77% |
| AWQ-W4-G64 | 4.31 | 5.64 | 91.28% | **59.20%** | 61.26% | 90.52% | 86.92% | 82.14% |
| BPDQ-W4-G128 | 4.63 | **4.95** | 92.65% | 55.40% | 62.88% | 90.70% | **87.21%** | **83.09%** |
| GPTQ-W3-G32 | 3.59 | 5.76 | **91.74%** | 51.00% | **63.05%** | 90.24% | 86.40% | **82.19%** |
| AWQ-W3-G32 | 3.59 | 5.57 | 90.90% | **58.60%** | 62.54% | **90.52%** | 86.63% | 82.01% |
| BPDQ-W3-G64 | 4.00 | **5.55** | 91.21% | 56.40% | 62.71% | **90.52%** | **86.73%** | 81.59% |
| GPTQ-W3-G64 | 3.30 | 6.04 | 90.07% | 50.80% | 59.98% | 90.12% | 86.22% | 81.55% |
| AWQ-W3-G64 | 3.30 | 5.85 | **90.75%** | **58.60%** | **63.74%** | 90.58% | 86.35% | 81.58% |
| BPDQ-W3-G128 | 3.50 | **5.73** | 90.67% | 56.60% | 62.80% | 90.52% | **86.36%** | **81.65%** |
| GPTQ-W2-G32 | 2.56 | 10.01 | 63.46% | 28.40% | 53.16% | 86.21% | 78.60% | 69.59% |
| AWQ-W2-G32 | 2.56 | 4.0E+7 | 0.00% | 0.00% | 41.47% | 68.75% | 58.09% | 56.94% |
| BPDQ-W2-G64 | 2.75 | **8.35** | 87.72% | 51.20% | 59.47% | 90.37% | 82.71% | 77.14% |
| GPTQ-W2-G64 | 2.28 | 12.47 | 40.49% | 14.40% | 41.89% | 79.79% | 74.69% | 62.18% |
| AWQ-W2-G64 | 2.28 | 1.6E+7 | 0.00% | 0.00% | 46.50% | 72.11% | 67.86% | 60.23% |
| BPDQ-W2-G128 | 2.38 | **8.66** | 86.13% | 47.60% | 60.75% | 90.06% | 82.20% | 76.73% |
| BPDQ-W2-G256 | 2.19 | 8.94 | 83.85% | 39.40% | 60.24% | 89.72% | 81.69% | 75.89% |

**Comparison with Bit-Plane and Vector Quantization Methods.** In addition to GPTQ and AWQ, the recent bit-plane method AnyBCQ (Park et al., 2025) and the vector quantization baseline VPTQ (Liu et al., 2024) are included.

In the 2-bit regime (Table 2), BPDQ, AnyBCQ, and VPTQ consistently outperform GPTQ and AWQ. While VPTQ achieves the highest accuracy, it incurs prohibitive quantization overhead ($\sim 40\times$ quantization time relative to GPTQ).

*Table 2.* Evaluation of BPDQ, GPTQ, AWQ, AnyBCQ (bit-plane method), and VPTQ (vector quantization) on Qwen2.5-7B.

| Model | SIZE(GB) | Wiki2 ↓ | GSM8K ↑ | MATH500 ↑ | ARC-C ↑ | BoolQ ↑ | HellaS ↑ | MMLU ↑ |
|---|---|---|---|---|---|---|---|---|
| *Qwen2.5-7B* | *14.19* | *9.42* | *75.97%* | *46.00%* | *55.29%* | *86.39%* | *80.44%* | *71.76%* |
| GPTQ-W4-G64 | 5.31 | 9.72 | 78.32% | 42.20% | 54.52% | **86.82%** | 80.00% | 71.16% |
| AWQ-W4-G64 | 5.31 | 10.35 | 78.29% | 45.20% | 55.12% | 86.24% | 79.93% | 71.08% |
| AnyBCQ-W4-G128 | 6.30 | 11.18 | 29.26% | 33.60% | 50.68% | 84.83% | 79.17% | 69.50% |
| VPTQ-W4 | 5.46 | **9.62** | **79.45%** | **47.00%** | 54.27% | 86.73% | 79.87% | **71.33%** |
| BPDQ-W4-G128 | 5.54 | 9.66 | 78.24% | 41.60% | **55.80%** | 86.42% | **80.03%** | 71.19% |
| GPTQ-W3-G32 | 4.77 | **10.04** | 72.48% | 44.40% | **54.78%** | 83.91% | **78.72%** | 68.97% |
| AWQ-W3-G32 | 4.76 | 10.70 | 57.16% | 44.60% | 51.71% | **86.36%** | 78.09% | 68.58% |
| AnyBCQ-W3-G64 | 5.82 | 12.24 | 26.61% | 25.60% | 50.34% | 82.39% | 77.21% | 66.99% |
| VPTQ-W3 | 4.51 | 10.32 | **78.17%** | **46.60%** | 51.28% | 85.96% | 78.17% | 69.78% |
| BPDQ-W3-G64 | 5.07 | 10.31 | 76.42% | 44.00% | 54.35% | 85.90% | 78.43% | **69.90%** |
| GPTQ-W3-G64 | 4.54 | **10.27** | 63.53% | 39.40% | 52.82% | 84.68% | **78.45%** | 67.53% |
| AWQ-W3-G64 | 4.54 | 11.28 | 65.58% | 38.00% | 50.17% | 84.95% | 77.27% | 67.23% |
| AnyBCQ-W3-G128 | 5.06 | 12.44 | 25.63% | 27.40% | 52.39% | 82.97% | 76.77% | 66.59% |
| BPDQ-W3-G128 | 4.69 | 10.55 | **71.27%** | **40.60%** | **54.27%** | **86.21%** | 77.92% | **69.53%** |
| GPTQ-W2-G32 | 3.98 | 21.66 | 0.38% | 3.00% | 34.04% | 65.02% | 66.12% | 37.12% |
| AWQ-W2-G32 | 3.98 | N/A | 2.43% | 0.00% | 34.64% | 45.99% | 48.98% | 28.30% |
| AnyBCQ-W2-G64 | 4.30 | 19.20 | 9.63% | 5.80% | 45.48% | 69.97% | 68.24% | 54.26% |
| VPTQ-W2 | 4.32 | **14.38** | **67.63%** | **33.40%** | 52.56% | 86.79% | **73.60%** | **65.81%** |
| BPDQ-W2-G64 | 4.12 | 15.09 | 44.50% | 13.60% | 48.29% | 85.50% | 69.98% | 57.51% |
| GPTQ-W2-G64 | 3.77 | 42.59 | 0.00% | 1.40% | 29.27% | 59.14% | 58.51% | 27.10% |
| AWQ-W2-G64 | 3.76 | N/A | 0.00% | 0.00% | 25.17% | 38.81% | 32.14% | 26.03% |
| AnyBCQ-W2-G128 | 3.92 | 22.57 | 4.47% | 5.80% | 44.54% | 77.52% | 65.83% | 48.54% |
| BPDQ-W2-G128 | 3.84 | **16.85** | **35.48%** | **10.40%** | **45.90%** | **84.62%** | **68.76%** | **57.46%** |

In contrast, BPDQ remains highly efficient (∼ 3×) with 10 iterations across all experiments. Furthermore, as a fellow bit-plane method, AnyBCQ also outperforms fixed-grid baselines in the extreme W2-G128 scenario. This confirms that the variable-grid structure offers stronger representation capabilities than fixed-grid data types at ultra-low bits. At 3-bit, BPDQ and VPTQ retain a marked lead on reasoning tasks (GSM8K, MATH500), whereas performance gaps narrow on general benchmarks. At 4-bit, most methods achieve high fidelity, with the exception of AnyBCQ, which still faces notable degradation on reasoning tasks.

### 4.3. Further Analysis of BPDQ

**System Efficiency Profile.** Experiments were conducted on a single NVIDIA H20 GPU. As shown in Table 3, BPDQ requires ∼3× the quantization time of GPTQ due to iteration (10 rounds), yet is far faster than VPTQ, which incurs an estimated ∼40× overhead. For inference, BPDQ utilizes the Look-Up Table (LUT) kernel (Park et al., 2022) adapted to support its bit-plane format, enabling efficient per-token decoding (Batch Size=1), which targets a real-time interactive generation scenario. In contrast, GPTQ utilizes optimized kernels (ExllamaV2 for W4, Torch/Triton for W3/W2). Overall, BPDQ achieves superior decoding latency in 2/3-bit regimes compared to GPTQ. While GPTQ-W4 also demonstrates competitive latency, it consumes higher VRAM (6.63 GB) due to ExllamaV2's pre-allocated

scratch buffers. VPTQ maintains consistent latency across bit-widths but suffers from prohibitive quantization costs.

**Activation Outlier Statistics.** We analyze activation outliers using 128 WikiText-2 sequences and report the results in Table 3. For outlier intensity, DiagR is defined as the max-to-median ratio per layer, and we report the 95th percentile (P95) across all layers. For outlier quantity, Cnt10 counts the number of channels exceeding $10\times$ the median, summed across all layers. Specifically, GPTQ-W2 exhibits severe suppression of outlier features ($\Delta$DiagR -32.89%, $\Delta$Cnt10 -23.61%). In contrast, VPTQ and BPDQ effectively retain these essential outliers under 2-bit quantization. While VPTQ employs expensive outlier protection, BPDQ inherently preserves outliers by extending the feasible set on a variable grid. Comparing the 2-bit results in Table 2 and Table 3, we observe a positive correlation between outlier preservation and downstream performance, consistent with (Lin et al., 2024; Gu et al., 2024; Xiong et al., 2025a).

**Long-Context Capabilities.** As illustrated in Figure 3, we evaluated the quantized models on a subset of LongBench covering retrieval (PassageRetrieval), summarization (Gov-Report, SAMSum), code completion (RepoBench-P), and classification (TREC). At 3-4 bits, all quantization methods show strong robustness on most tasks, maintaining performance generally comparable to the baseline. A significant challenge arises at 2-bit, particularly in the retrieval task,

*Table 3.* System efficiency profile and activation outlier statistics on Qwen2.5-7B.

| Model | Efficiency Profile | | | Outlier Statistics | | | |
|---|---|---|---|---|---|---|---|
| | Cost (min) | VRAM (GB) | Latency (ms) | DiagR (P95) | △DiagR | Cnt10 | △Cnt10 |
| Qwen2.5-7B | N/A | 14.19 | 14.42 | 3.01E4 | N/A | 4.32E4 | N/A |
| GPTQ-W4-G64 | 16 | 6.63 | 18.74 | 3.28E4 | +8.97% | 4.28E4 | -0.93% |
| VPTQ-W4 | 4×160 [†] | 5.46 | 20.07 | 2.83E4 | -5.98% | 4.30E4 | -0.46% |
| BPDQ-W4-G128 | 47 | 5.55 | 18.20 | 2.95E4 | -1.99% | 4.28E4 | -0.93% |
| GPTQ-W3-G32 | 16 | 4.54 | 47.67 | 2.82E4 | -6.31% | 4.12E4 | -4.63% |
| VPTQ-W3 | 4×160 [†] | 4.51 | 17.34 | 2.59E4 | -13.95% | 4.38E4 | +1.39% |
| BPDQ-W3-G64 | 40 | 4.69 | 18.21 | 2.96E4 | -1.66% | 4.29E4 | -0.69% |
| GPTQ-W2-G32 | 17 | 3.77 | 33.91 | 2.02E4 | -32.89% | 3.30E4 | -23.61% |
| VPTQ-W2 | 4×170 [†] | 4.32 | 18.24 | 2.68E4 | -10.96% | 4.44E4 | +2.78% |
| BPDQ-W2-G64 | 40 | 3.86 | 18.09 | 2.86E4 | -4.98% | 4.24E4 | -1.85% |

[†] VPTQ costs from the paper require 4 GPUs and ∼10× time relative to the single-GPU GPTQ baseline.

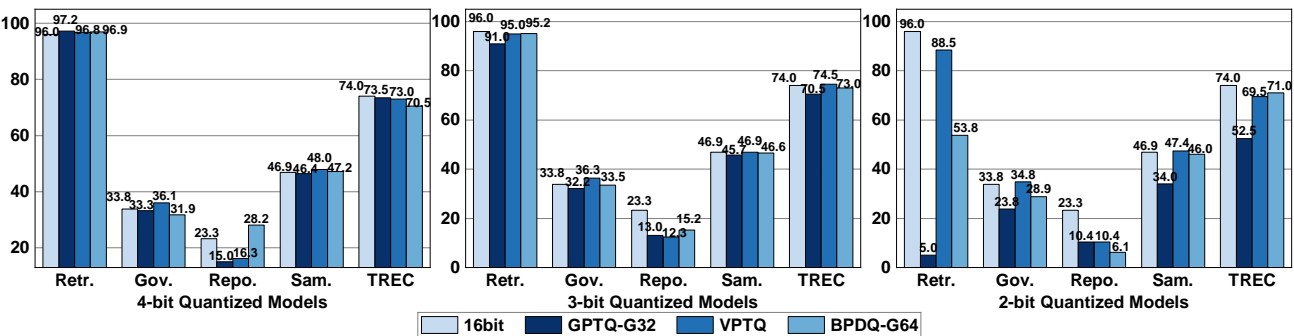

*Figure 3.* LongBench performance comparison on Qwen2.5-7B.

which acts as a stress test for long-range dependency (Xiong et al., 2025b). GPTQ suffers severe degradation (score drops to 4.98%), indicating the loss of retrieval capabilities. In contrast, BPDQ sustains the performance at 53.75%, whereas VPTQ achieves higher resilience but at the cost of prohibitive quantization overhead. Furthermore, in summarization and classification tasks, BPDQ performs competitively with the baseline under such extreme compression.

## 5. Conclusion

In this paper, we present Bit-Plane Decomposition Quantization (BPDQ) to relax the constraint of shape-invariant grids that hampers optimization-based PTQ in low-bit regimes. Specifically, BPDQ constructs a variable quantization grid via bit-plane decomposition, which theoretically expands the feasible solution set and allows for a rigorous refinement process within the Hessian-induced geometry. Consequently, BPDQ unlocks high-fidelity 2-bit inference for 72B models on consumer-grade GPUs. By relaxing the rigidity of the quantization grid while maintaining a hardware-friendly format, BPDQ offers a promising direction for extreme model compression and efficient deployment.

## 6. Limitations and Future Work

**Fidelity Gap and Enhancements.** While BPDQ achieves strong performance, a fidelity gap remains compared to vector quantization, which often has high overhead and limited hardware support. Future work could address this by incorporating rotation techniques (Ashkboos et al., 2024), or by integrating enhanced sequential solvers like Qronos, to maximize the potential of the optimization framework.

**Hardware Efficiency on FPGA/ASIC.** The binary nature of bit-planes ($\{0, 1\}$) is inherently suitable for FPGA or ASIC deployment (Zeng et al.; Hong et al., 2022). This suits custom hardware, as it allows replacing expensive floating-point multiplications with simple additions, significantly improving energy and area efficiency.

**Mixed- and Multi-Precision.** BPDQ's unified basis inherently surpasses conventional mixed-precision schemes. Instead of requiring hardware support for diverse data types, BPDQ achieves mixed precision simply by allocating more or fewer bit-planes. Furthermore, this structure naturally supports multi-precision serving (Park et al., 2025), enabling dynamic accuracy-latency trade-offs by serving multiple precisions from a single on-device model.

## Impact Statement

This paper presents work whose goal is to advance the field of Machine Learning. There are many potential societal consequences of our work, none which we feel must be specifically highlighted here.

## Acknowledgments

This work was supported in part by the Theme-based Research Scheme (TRS) project T45-701/22-R of the Research Grants Council of Hong Kong, and in part by the AVNET-HKU Emerging Microelectronics and Ubiquitous Systems (EMUS) Lab. The work of Long Shi was supported by the National Natural Science Foundation of China under Grant 62201475 and Sichuan Science and Technology Program under Grant 2024NSFSC1436. This work was also supported in part by grants from the Research Grants Council of the Hong Kong Special Administrative Region, China (Project No. T41-517/25-N and 15228325 )

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

# A. Analysis of the Variable Grid

## A.1. Optimization-based PTQ as an H-Metric Projection

Consider a weight vector $\mathbf{w} \in \mathbb{R}^g$ within a quantization group of size $g$. Let $\mathbf{H} \in \mathbb{R}^{g \times g}$ denote the corresponding Hessian matrix. We define the Hessian-induced norm as $\|\mathbf{e}\|_{\mathbf{H}} = \sqrt{\mathbf{e}^\top \mathbf{H} \mathbf{e}}$. Optimization-based PTQ determines a quantized vector $\widehat{\mathbf{w}}$ from a feasible set $\mathcal{Q} \subset \mathbb{R}^g$ that minimizes the output discrepancy. This is equivalent to finding the projection of $\mathbf{w}$ onto $\mathcal{Q}$ under the $\mathbf{H}$-metric:

$$\widehat{\mathbf{w}} \ = \ \Pi_{\mathcal{Q}}^{(\mathbf{H})}(\mathbf{w}) \ = \ \underset{\widetilde{\mathbf{w}} \in \mathcal{Q}}{\arg\min} \|\mathbf{w} - \widetilde{\mathbf{w}}\|_{\mathbf{H}}^2 \ = \ \underset{\widetilde{\mathbf{w}} \in \mathcal{Q}}{\arg\min} (\mathbf{w} - \widetilde{\mathbf{w}})^\top \mathbf{H} (\mathbf{w} - \widetilde{\mathbf{w}}). \tag{10}$$

Eq. (10) establishes that optimization-based PTQ is a nearest-point projection problem. Consequently, quantization quality is restricted by the geometric richness of the feasible set $\mathcal{Q}$. In the low-bit regime (e.g., 2-3 bits), fidelity degradation stems not from a failure of the objective, but from the rigidity of the feasible set $\mathcal{Q}$ induced by a shape-invariant grid.

## A.2. Feasible Set Comparison at 2-Bit Precision

The distinction between fixed and variable grids lies in the geometric degrees of freedom defining the quantization levels within each group. A fixed grid constrains the levels to a shape-invariant template governed by a scaling factor, restricting the solution to a lower-dimensional manifold. In contrast, BPDQ constructs the grid using independent scalar coefficients for each group, decoupling the quantization intervals and expanding the feasible set to a higher-dimensional geometry.

**Fixed Grid (Rigid Template).** Given a canonical template $\mathbf{t} = [t_0, t_1, t_2, t_3]^\top \in \mathbb{R}^4$ (e.g., $[0, 1, 2, 3]^\top$ for UINT2), a fixed grid restricts the quantization levels $\mathbf{q} \in \mathbb{R}^4$ to a scaling factor of this template. For a group scale $s \in \mathbb{R}$:

$$\mathcal{Q}_{\text{fix}}(s) \ = \ s \cdot \{t_0, t_1, t_2, t_3\}, \tag{11}$$

where $s$ varies across groups, but the relative ratios between levels (e.g., $q_2/q_1 = (s \cdot t_2)/(s \cdot t_1) = t_2/t_1, t_1 \neq 0$) remain frozen. This rigidity restricts the feasible level vectors to a one-dimensional ray in $\mathbb{R}^4$.

**Variable Grid (Adaptive Geometry).** BPDQ constructs the grid via two bit-planes weighted by coefficients $c_1, c_2 \in \mathbb{R}$:

$$\mathcal{Q}_{\text{var}}(c_1, c_2) \ = \ \{0, \ c_1, \ c_2, \ c_1 + c_2\}, \tag{12}$$

where $c_1$ and $c_2$ are independent coefficients determined per group, enabling dynamic relative ratios (i.e., $c_2/c_1$ is flexible, $c_1 \neq 0$). This flexibility expands the feasible level vectors to a two-dimensional plane in $\mathbb{R}^4$.

**Proposition 1: Strict Inclusion of Uniform Grids.** *The feasible set of BPDQ strictly contains the feasible set of standard UINT2 grids.*

*Proof.* Without loss of generality, factoring out the group-wise bias (available to both schemes), we consider the canonical zero-based template $\mathbf{t}_{\text{uni}} = \{0, 1, 2, 3\}$. Accordingly, any uniform grid is essentially a scaled instance of this template, given by $s \cdot \{0, 1, 2, 3\} = \{0, s, 2s, 3s\}$. To show inclusion ($\mathcal{Q}_{\text{uniform}} \subset \mathcal{Q}_{\text{BPDQ}}$), we set the coefficients $c_1 = s$ and $c_2 = 2s$:

$$\mathcal{Q}_{\text{var}}(s, 2s) = \{0, \ s, \ 2s, \ s + 2s\} = \{0, s, 2s, 3s\} \equiv \mathcal{Q}_{\text{fix}}^{\text{uni}}(s). \tag{13}$$

This confirms that BPDQ can exactly reproduce any UINT2 grid. Furthermore, the inclusion is strict because BPDQ admits non-uniform spacings (e.g., when $c_2 = 10c_1$) that no linear scale $s$ can represent. Consequently, the quantization error of BPDQ is upper-bounded by that of UINT2:

$$\min_{\mathbf{q} \in \mathcal{Q}_{\text{var}}} \|\mathbf{w} - \mathbf{q}\|_{\mathbf{H}}^2 \leq \min_{\mathbf{q} \in \mathcal{Q}_{\text{fix}}^{\text{uni}}} \|\mathbf{w} - \mathbf{q}\|_{\mathbf{H}}^2. \tag{14}$$

$\square$

**Proposition 2: Strict Error Reduction via Variable-Grid Expressivity.** *Compared to shape-invariant fixed-template quantization grids parameterized only by a per-group bias $c_0$ and scale $s$, 2-bit BPDQ induces additional degrees of freedom and can realize feasible points unattainable by rigid templates. Consequently, there exists a non-empty open set of weight vectors $\mathcal{U} \subset \mathbb{R}^g$ where BPDQ achieves strictly lower quantization error.*

*Proof.* Assume group size $g \geq 3$ and Hessian $\mathbf{H} \succ 0$. Define the feasible sets for the fixed grid ($\mathcal{S}_{\mathrm{fix}}$) and BPDQ ($\mathcal{S}_{\mathrm{var}}$):

$$\mathcal{S}_{\mathrm{fix}} = \{c_0 \mathbf{1} + s\mathbf{z} \mid c_0, s \in \mathbb{R}, \mathbf{z} \in \{t_0, \ldots, t_3\}^g\}, \tag{15}$$

$$\mathcal{S}_{\mathrm{var}} = \{c_0 \mathbf{1} + c_1 \mathbf{b}_1 + c_2 \mathbf{b}_2 \mid c_0, c_1, c_2 \in \mathbb{R}, \mathbf{b}_1, \mathbf{b}_2 \in \{0, 1\}^g\}. \tag{16}$$

For any specific pattern $\mathbf{z}$ (or bit-planes $\mathbf{b}_1, \mathbf{b}_2$), the generated vectors form an affine subspace. Since the number of patterns is finite ($4^g$), both $\mathcal{S}_{\mathrm{fix}}$ and $\mathcal{S}_{\mathrm{var}}$ are finite unions of affine subspaces, and are thus closed sets.

**Construction of $\mathbf{v}^*$:** Define the finite set of difference ratios for $\mathbf{t} = [t_0, t_1, t_2, t_3]^\top \in \mathbb{R}^4$:

$$\mathcal{R}_\Delta(\mathbf{t}) = \left\{ \frac{t_i - t_j}{t_i - t_k} \;\middle|\; t_i, t_j, t_k \in \{t_0, \ldots, t_3\}, t_i \neq t_j, t_i \neq t_k, t_j \neq t_k \right\}. \tag{17}$$

For any $\mathbf{q} \in \mathcal{S}_{\mathrm{fix}}$, if $\mathbf{q}$ attains three distinct values $x, y, z$ across three coordinates (so $t_i \neq t_j, t_i \neq t_k$ and $t_j \neq t_k$ when $s \neq 0$), they must satisfy $x = c_0 + st_i$, $y = c_0 + st_j$, $z = c_0 + st_k$. The bias and scale cancel out in the difference ratio: $(x - y)/(x - z) = (t_i - t_j)/(t_i - t_k) \in \mathcal{R}_\Delta(\mathbf{t})$. In contrast, BPDQ can generate a vector $\mathbf{v}^*$ containing values $\{c_0, c_0 + c_1, c_0 + c_2\}$ by selecting bit-planes such that three coordinates take patterns $(0, 0), (1, 0), (0, 1)$. Let these three values be $x = c_0, y = c_0 + c_1, z = c_0 + c_2$. The difference ratio becomes:

$$\frac{x - y}{x - z} = \frac{c_0 - (c_0 + c_1)}{c_0 - (c_0 + c_2)} = \frac{c_1}{c_2}. \tag{18}$$

Since $\mathcal{R}_\Delta(\mathbf{t})$ is a finite set, we can choose $c_1, c_2 \in \mathbb{R}$ such that the ratio $c_1/c_2 \notin \mathcal{R}_\Delta(\mathbf{t})$ and the resulting values $x, y, z$ are distinct. Suppose for contradiction that $\mathbf{v}^* \in \mathcal{S}_{\mathrm{fix}}$. As $\mathbf{v}^*$ attains the distinct values $x, y, z$ at the chosen coordinates, the fixed-grid constraint would necessitate that their difference ratio falls within $\mathcal{R}_\Delta(\mathbf{t})$, which contradicts our construction. Thus, $\mathbf{v}^* \in \mathcal{S}_{\mathrm{var}}$ but $\mathbf{v}^* \notin \mathcal{S}_{\mathrm{fix}}$.

**Strict Inequality over an Open Set:** Consider a weight vector $\mathbf{w} = \mathbf{v}^*$, for which the BPDQ error is zero (i.e., $F_{\mathrm{var}}(\mathbf{v}^*) = 0$). Since $\mathbf{H} \succ 0$ induces a norm equivalent to the Euclidean norm on $\mathbb{R}^g$, and $\mathcal{S}_{\mathrm{fix}}$ is a closed set with $\mathbf{v}^* \notin \mathcal{S}_{\mathrm{fix}}$, the distance is strictly positive:

$$F_{\mathrm{fix}}(\mathbf{v}^*) = \inf_{\mathbf{q} \in \mathcal{S}_{\mathrm{fix}}} \|\mathbf{v}^* - \mathbf{q}\|_{\mathbf{H}}^2 = \delta > 0. \tag{19}$$

Define the error difference function $f(\mathbf{w}) = F_{\mathrm{fix}}(\mathbf{w}) - F_{\mathrm{var}}(\mathbf{w})$. Since $\mathcal{S}_{\mathrm{fix}}$ and $\mathcal{S}_{\mathrm{var}}$ are closed sets, their respective squared-distance functions $F_{\mathrm{fix}}(\mathbf{w})$ and $F_{\mathrm{var}}(\mathbf{w})$ are continuous. Specifically, the distance-to-set function $d(\mathbf{w}, \mathcal{S})$ is 1-Lipschitz under $\|\cdot\|_{\mathbf{H}}$, and squaring preserves continuity. Therefore, $f(\mathbf{w})$ is continuous, so there exists an open neighborhood $\mathcal{U}$ around $\mathbf{v}^*$ where $f(\mathbf{w}) > 0$ (i.e., $F_{\mathrm{fix}}(\mathbf{w}) > F_{\mathrm{var}}(\mathbf{w})$). This confirms the existence of a strict inequality over an open set. $\square$

**Remark: Geometric Degrees of Freedom.** Proposition 1 establishes strictly nested feasibility for uniform grids ($\mathcal{S}_{\mathrm{uni}} \subsetneq \mathcal{S}_{\mathrm{BPDQ}}$), and Proposition 2 addresses general fixed templates. Geometrically, a specific choice of bit-planes ($\mathbf{b}_1, \mathbf{b}_2$) in BPDQ yields an affine subspace of dimension up to 3 (spanning $\mathbf{1}, \mathbf{b}_1, \mathbf{b}_2$), whereas fixed templates yield subspaces of dimension at most 2 (spanning $\mathbf{1}, \mathbf{z}$). In the generic case where ($\mathbf{1}, \mathbf{b}_1, \mathbf{b}_2$) are linearly independent, this provides an additional coefficient degree of freedom (3 parameters ($c_0, c_1, c_2$) vs. 2 parameters ($c_0, s$)). ***Crucially, the fixed grid enforces global rigidity across all groups, whereas the variable grid enables adaptability tailored for each group.***

## B. Consistency in Hessian-Induced Geometry

### B.1. Consistency of Coefficient Fitting

**Proposition.** *The coefficient fitting objective in Eq. (6) is theoretically equivalent to minimizing the local contribution to the optimization objective in Eq. (2) corresponding to the current group, under the Hessian-induced geometry defined by the upper-triangular Cholesky factor.*

*Proof.* The optimization objective minimizes the output reconstruction error defined by the Hessian $\mathbf{H}$:

$$\mathcal{L} = \mathrm{tr}\left((\mathbf{W} - \widehat{\mathbf{W}})\mathbf{H}(\mathbf{W} - \widehat{\mathbf{W}})^\top\right). \tag{20}$$

Substituting the Cholesky factorization of the inverse Hessian $\mathbf{H}^{-1} = \mathbf{U}^\top \mathbf{U}$ (i.e., $\mathbf{H} = \mathbf{U}^{-1}\mathbf{U}^{-\top}$), we rewrite the objective using the definition of the Frobenius norm:

$$\mathcal{L} = \left\|(\mathbf{W} - \widehat{\mathbf{W}})\mathbf{U}^{-1}\right\|_F^2 = \left\|\mathbf{U}^{-\top}(\mathbf{W} - \widehat{\mathbf{W}})^\top\right\|_F^2. \tag{21}$$

This reveals that the error measures the magnitude of the weight residual projected onto the geometry defined by the inverse Cholesky factor. When determining the coefficients for a column group $\mathbf{W}_{:,s:(s+g)}$, we consider the corresponding local block $\mathbf{U}_{\mathrm{loc}} = \mathbf{U}_{s:(s+g),s:(s+g)} \in \mathbb{R}^{g \times g}$. Accordingly, the local contribution to the error is:

$$\mathcal{L}_{\mathrm{loc}} = \left\|\mathbf{U}_{\mathrm{loc}}^{-\top}(\mathbf{W}_{:,s:(s+g)} - \widehat{\mathbf{W}}_{:,s:(s+g)})^\top\right\|_F^2. \tag{22}$$

Since the Frobenius norm decomposes row-wise, we minimize the error for each row $r$ independently. In BPDQ, the quantized row vector is parameterized as $\widehat{\mathbf{W}}_{r,s:(s+g)}^\top = \mathbf{B}_r c_r$, where $c_r \in \mathbb{R}^{k+1}$ is the coefficient vector. Substituting this parameterization and the original weight vector $\mathbf{W}_{r,s:(s+g)}^\top$ into the row-wise objective yields:

$$\underset{c_r \in \mathbb{R}^{k+1}}{\arg\min} \left\|\mathbf{U}_{\mathrm{loc}}^{-\top}(\mathbf{B}_r c_r - \mathbf{W}_{r,s:(s+g)}^\top)\right\|_2^2. \tag{23}$$

This matches exactly the weighted least-squares problem defined in Eq. (6). Thus, solving Eq. (6) minimizes the optimization objective over the group coefficients within the Hessian-induced geometry. In implementation, a damping factor $\alpha = 10^{-4}$ is applied to the diagonal for numerical stability (omitted in the derivation for brevity). $\square$

### B.2. Consistency of Bit-Plane Update

**Proposition.** *During the bit-plane update (with fixed coefficients), the element-wise enumeration within column $l$ minimizes the column-wise error term of the optimization objective in Eq. (2). Under the error propagation mechanism, this element-wise Euclidean projection ensures consistency with the Hessian-induced geometry.*

*Proof.* As derived in Eq. (21), the objective is equivalent to minimizing the projected residual norm $\mathcal{L} = \|(\mathbf{W} - \widehat{\mathbf{W}})\mathbf{U}^{-1}\|_F^2$. The error propagation mechanism in Eq. (4) establishes the relationship $(\mathbf{W} - \widehat{\mathbf{W}}) = \mathbf{E}\mathbf{U}$, where $\mathbf{E}$ is the matrix collecting the error coordinates $\mathbf{E}_{:,l}$ defined in Eq. (3). Substituting this relationship into the objective $\mathcal{L}$:

$$\mathcal{L} = \left\|(\mathbf{E}\mathbf{U})\mathbf{U}^{-1}\right\|_F^2 = \left\|\mathbf{E}\right\|_F^2 = \sum_{l=1}^{d_{\mathrm{in}}} \|\mathbf{E}_{:,l}\|_2^2. \tag{24}$$

For the $l$-th column, the objective reduces to minimizing the error term $\|\mathbf{E}_{:,l}\|_2^2$ conditioned on the current propagation state. Based on the definition in Eq. (3), the error coordinate for the current working column $\mathbf{W}_{:,l}'$ is:

$$\mathbf{E}_{:,l} = \frac{\mathbf{W}_{:,l}' - \widehat{\mathbf{W}}_{:,l}}{\mathbf{U}_{l,l}}. \tag{25}$$

Assuming $\mathbf{H} \succ 0$, the diagonal element $\mathbf{U}_{l,l}$ is a strictly positive scalar constant independent of the quantization choice $\widehat{\mathbf{W}}_{:,l}$. Therefore, minimizing the column-wise error coordinate $\|\mathbf{E}_{:,l}\|_2^2$ is strictly equivalent to minimizing the Euclidean distance in the weight space:

$$\widehat{\mathbf{W}}_{:,l} = \underset{\widetilde{\mathbf{W}}_{:,l}}{\operatorname{argmin}} \left\| \frac{\mathbf{W}'_{:,l} - \widetilde{\mathbf{W}}_{:,l}}{\mathbf{U}_{l,l}} \right\|_2^2 = \underset{\widetilde{\mathbf{W}}_{:,l}}{\operatorname{argmin}} \|\mathbf{W}'_{:,l} - \widetilde{\mathbf{W}}_{:,l}\|_2^2. \tag{26}$$

Since the group-wise coefficients are fixed, this problem decouples into $d_{\text{out}}$ independent scalar nearest-neighbor projections. For each coordinate $r$ of the column, the value $\widetilde{\mathbf{W}}_{r,l}$ must be selected from the set of values $v_r(\mathbf{b})$ generated by the bit vectors $\mathbf{b} \in \{0,1\}^k$ following Eq. (7). Thus, the problem simplifies to finding the optimal bit vector $\mathbf{b}^*$ for each coordinate:

$$\mathbf{b}^* = \underset{\mathbf{b} \in \{0,1\}^k}{\operatorname{argmin}} (\mathbf{W}'_{r,l} - v_r(\mathbf{b}))^2. \tag{27}$$

This matches Eq. (8), confirming that the element-wise Euclidean nearest-neighbor projection minimizes the column-wise error, and maintains consistency with the Hessian-induced geometry. $\qquad\square$

### B.3. Consistency of Delta Correction

**Proposition.** *The delta correction in Eq. (9) preserves the error-propagation consistency following coefficient refitting.*

*Proof.* We partition the global index space into the current group columns $\{s : (s + g)\}$ and the tail columns $\{(s + g) : d_{\text{in}}\}$. Recall the propagation invariant $(\mathbf{W} - \widehat{\mathbf{W}}) = \mathbf{E}\mathbf{U}$ from Appendix B.2 and let $\mathbf{U}_{\text{loc}} = \mathbf{U}_{s:(s+g),s:(s+g)} \in \mathbb{R}^{g \times g}$.

To preserve the *local consistency* within the group columns, we utilize the upper triangular structure to decompose the residual:

$$\mathbf{W}_{:,s:(s+g)} - \widehat{\mathbf{W}}_{:,s:(s+g)} = (\mathbf{E}\mathbf{U})_{:,s:(s+g)} = \mathbf{E}_{:,:s}\mathbf{U}_{:s,s:(s+g)} + \mathbf{E}_{:,s:(s+g)}\mathbf{U}_{\text{loc}}. \tag{28}$$

By rearranging Eq. (28), we isolate the components that remain constant during coefficient refitting (i.e., original weights and historical errors):

$$\underbrace{\mathbf{W}_{:,s:(s+g)} - \mathbf{E}_{:,:s}\mathbf{U}_{:s,s:(s+g)}}_{\text{Constant}} = \widehat{\mathbf{W}}_{:,s:(s+g)} + \mathbf{E}_{:,s:(s+g)}\mathbf{U}_{\text{loc}}. \tag{29}$$

Since the left side of Eq. (29) is constant, the right side must be equal for the bit-plane update state (old) and the refitted state (new). Subtracting the expression for the new state from the old state:

$$(\mathbf{E}^{\text{new}}_{:,s:(s+g)} - \mathbf{E}^{\text{old}}_{:,s:(s+g)})\mathbf{U}_{\text{loc}} = \widehat{\mathbf{W}}_{\text{old}} - \widehat{\mathbf{W}}_{\text{new}}. \tag{30}$$

This strictly derives the delta correction $\Delta\mathbf{E}\,\mathbf{U}_{\text{loc}} = \widehat{\mathbf{W}}_{\text{old}} - \widehat{\mathbf{W}}_{\text{new}}$ in Eq. (9).

To preserve the *tail consistency*, we expand the tail working weights $\mathbf{W}'_{:,(s+g):}$ based on the error propagation in Eq. (4) and the decomposition analogous to Eq. (28):

$$\mathbf{W}'_{:,(s+g):} = \underbrace{\mathbf{W}_{:,(s+g):} - \sum_{j<s} \mathbf{E}_{:,j}\mathbf{U}_{j,(s+g):}}_{\text{History (Constant)}} - \underbrace{\mathbf{E}_{:,s:(s+g)}\mathbf{U}_{s:(s+g),(s+g):}}_{\text{Current Group (Varying)}}. \tag{31}$$

The first term accounts for the original weights and errors from preceding groups ($j < s$), which remain constant during the current group's coefficient refitting. The change in the working weights is derived by differencing Eq. (31) between the new and old states:

$$\mathbf{W}'^{\text{new}}_{:,(s+g):} - \mathbf{W}'^{\text{old}}_{:,(s+g):} = -(\mathbf{E}^{\text{new}}_{:,s:(s+g)} - \mathbf{E}^{\text{old}}_{:,s:(s+g)})\mathbf{U}_{s:(s+g),(s+g):} = -\Delta\mathbf{E}\,\mathbf{U}_{s:(s+g),(s+g):}. \tag{32}$$

Thus, applying the update $\Delta\mathbf{E}$ exactly synchronizes the propagation state for the tail columns. $\qquad\square$

## C. Additional Evaluation Results

*Table 4.* Evaluation results of Qwen3-0.6B, Ministral3-3B, and Qwen3-4B across seven benchmarks. Best and second-best results are highlighted in **bold** and underlined, respectively.

| Model | BPW | Wiki2 ↓ | GSM8K ↑ | MATH500 ↑ | ARC-C ↑ | BoolQ ↑ | HellaS ↑ | MMLU ↑ |
|---|---|---|---|---|---|---|---|---|
| *Qwen3-0.6B* | *16* | *26.09* | *41.02%* | *28.60%* | *33.70%* | *63.82%* | *47.30%* | *40.39%* |
| GPTQ-W4-G64 | 4.31 | 28.61 | 30.10% | **22.40%** | 30.80% | 64.71% | **46.33%** | 34.05% |
| AWQ-W4-G64 | 4.31 | 29.53 | **36.24%** | **22.40%** | 31.91% | **66.91%** | 45.57% | **41.57%** |
| BPDQ-W4-G128 | 4.63 | **28.58** | 31.54% | 18.60% | **32.08%** | 57.43% | 45.88% | 29.65% |
| GPTQ-W3-G32 | 3.59 | 35.15 | **18.95%** | 6.40% | 31.23% | **66.70%** | 42.86% | 25.35% |
| AWQ-W3-G32 | 3.59 | 37.43 | 9.63% | **6.80%** | 29.10% | 49.79% | 42.67% | 36.13% |
| BPDQ-W3-G64 | 4.00 | **34.38** | 14.03% | 6.00% | 27.90% | 55.87% | **43.06%** | **36.88%** |
| GPTQ-W3-G64 | 3.30 | **38.14** | 5.69% | 4.00% | **29.86%** | 56.73% | 41.90% | 30.49% |
| AWQ-W3-G64 | 3.30 | 44.48 | 3.64% | **5.00%** | 26.71% | 58.50% | 40.81% | **34.90%** |
| BPDQ-W3-G128 | 3.50 | 38.40 | **7.43%** | 4.60% | 29.52% | **64.13%** | **42.07%** | 28.03% |
| GPTQ-W2-G32 | 2.56 | 3.7E+2 | 0.00% | 1.80% | 23.29% | 40.49% | 27.95% | 24.61% |
| AWQ-W2-G32 | 2.56 | 5.6E+6 | 0.00% | 0.00% | **26.28%** | 47.19% | 27.05% | **24.79%** |
| BPDQ-W2-G64 | 2.75 | **1.2E+2** | **0.15%** | **2.40%** | 23.55% | **62.29%** | **34.09%** | 23.35% |
| GPTQ-W2-G64 | 2.28 | 1.2E+3 | 0.00% | **1.60%** | 22.70% | 39.11% | 26.04% | 24.73% |
| AWQ-W2-G64 | 2.28 | 5.8E+7 | 0.00% | 0.20% | **27.30%** | 51.25% | 26.07% | **25.92%** |
| BPDQ-W2-G128 | 2.38 | **1.3E+2** | **0.30%** | **1.60%** | 21.93% | **60.95%** | **32.66%** | 25.36% |
| *Ministral-3-3B* | *16* | *11.70* | *73.16%* | *40.00%* | *60.41%* | *84.10%* | *73.45%* | *67.41%* |
| GPTQ-W4-G64 | 4.31 | **12.09** | **70.43%** | 41.40% | **59.56%** | 83.33% | **72.88%** | 65.25% |
| AWQ-W4-G64 | 4.31 | 12.22 | 70.05% | **42.60%** | **59.56%** | 83.06% | 72.50% | 65.79% |
| BPDQ-W4-G128 | 4.63 | 12.10 | 70.13% | 37.40% | 58.11% | **84.13%** | 72.78% | **66.09%** |
| GPTQ-W3-G32 | 3.59 | 13.30 | 65.05% | 30.60% | 54.86% | 82.51% | **71.19%** | 62.74% |
| AWQ-W3-G32 | 3.59 | 13.83 | 61.26% | 30.00% | 54.69% | 79.60% | 69.93% | 63.35% |
| BPDQ-W3-G64 | 4.00 | **13.16** | **65.43%** | **34.80%** | **56.40%** | **83.33%** | 70.46% | **63.39%** |
| GPTQ-W3-G64 | 3.30 | 13.90 | **61.79%** | 28.00% | **55.55%** | 81.01% | **70.55%** | 61.87% |
| AWQ-W3-G64 | 3.30 | 14.41 | 54.21% | 29.60% | **55.55%** | 80.83% | 68.87% | **62.41%** |
| BPDQ-W3-G128 | 3.50 | **13.52** | 58.83% | **32.80%** | 53.75% | **81.44%** | 70.14% | 61.95% |
| GPTQ-W2-G32 | 2.56 | 37.76 | 1.06% | 3.00% | 26.71% | 49.69% | 46.40% | 25.23% |
| AWQ-W2-G32 | 2.56 | 9.3E+5 | 0.00% | 0.00% | 25.09% | 51.35% | 35.90% | 24.18% |
| BPDQ-W2-G64 | 2.75 | **21.01** | **21.99%** | **5.60%** | **41.47%** | **72.08%** | **59.31%** | **49.67%** |
| GPTQ-W2-G64 | 2.28 | 69.48 | 0.61% | 2.40% | 26.54% | 56.18% | 37.70% | 23.68% |
| AWQ-W2-G64 | 2.28 | 1.8E+6 | 0.00% | 0.00% | 24.57% | 38.01% | 27.43% | 25.51% |
| BPDQ-W2-G128 | 2.38 | **23.46** | **17.36%** | **6.20%** | **40.27%** | **78.38%** | **57.23%** | **45.24%** |
| *Qwen3-4B* | *16* | *13.07* | *85.75%* | *52.60%* | *58.62%* | *84.74%* | *69.08%* | *70.57%* |
| GPTQ-W4-G64 | 4.31 | **13.28** | **85.44%** | 50.00% | 57.76% | 84.28% | 68.60% | 69.29% |
| AWQ-W4-G64 | 4.31 | 13.60 | **85.44%** | 48.80% | 57.51% | 83.88% | 68.58% | 69.35% |
| BPDQ-W4-G128 | 4.63 | 13.57 | 81.50% | **51.20%** | **59.22%** | **84.86%** | **68.87%** | **69.80%** |
| GPTQ-W3-G32 | 3.59 | **14.05** | 70.13% | 44.80% | 56.23% | 84.22% | **67.19%** | 66.19% |
| AWQ-W3-G32 | 3.59 | 14.95 | **79.91%** | **47.80%** | 54.01% | 82.14% | 65.52% | 66.71% |
| BPDQ-W3-G64 | 4.00 | 14.67 | 77.71% | 46.40% | **57.42%** | **84.56%** | 66.44% | **67.06%** |
| GPTQ-W3-G64 | 3.30 | **14.53** | 73.39% | 38.00% | 55.03% | 83.15% | 66.24% | 65.96% |
| AWQ-W3-G64 | 3.30 | 16.02 | **79.38%** | **43.80%** | 49.15% | 81.87% | 64.32% | 64.46% |
| BPDQ-W3-G128 | 3.50 | 14.79 | 61.87% | 40.60% | **55.12%** | **83.52%** | **65.81%** | **66.70%** |
| GPTQ-W2-G32 | 2.56 | 23.37 | 0.61% | 1.80% | 33.96% | 52.81% | 52.04% | 27.45% |
| AWQ-W2-G32 | 2.56 | 1.8E+8 | 0.30% | 0.00% | 33.45% | 55.81% | 45.85% | 36.03% |
| BPDQ-W2-G64 | 2.75 | **21.40** | **34.19%** | **8.80%** | **42.92%** | **78.78%** | **56.81%** | **53.67%** |
| GPTQ-W2-G64 | 2.28 | 34.80 | 0.00% | 1.20% | 28.24% | 50.15% | 44.36% | 24.47% |
| AWQ-W2-G64 | 2.28 | 8.5E+8 | 0.00% | 0.00% | 26.62% | 46.36% | 28.97% | 25.42% |
| BPDQ-W2-G128 | 2.38 | **23.93** | **24.87%** | **4.80%** | **40.19%** | **80.61%** | **55.09%** | **52.49%** |

*Table 5.* Evaluation results of Qwen3-8B and Qwen3-14B across seven benchmarks.

| Model | BPW | Wiki2 ↓ | GSM8K ↑ | MATH500 ↑ | ARC-C ↑ | BoolQ ↑ | HellaS ↑ | MMLU ↑ |
|---|---|---|---|---|---|---|---|---|
| *Qwen3-8B* | *16* | *12.22* | *87.11%* | *53.00%* | *56.74%* | *86.61%* | *74.90%* | *73.02%* |
| GPTQ-W4-G64 | 4.31 | **12.52** | **87.04%** | **52.40%** | 55.38% | 86.24% | **74.57%** | 72.33% |
| AWQ-W4-G64 | 4.31 | 12.78 | 86.28% | 50.60% | **55.81%** | 86.38% | 73.55% | 72.18% |
| BPDQ-W4-G128 | 4.63 | 12.66 | 86.43% | 48.80% | 55.72% | **86.85%** | 74.13% | **72.39%** |
| GPTQ-W3-G32 | 3.59 | **12.97** | 82.64% | 47.60% | 53.24% | 85.26% | **73.05%** | **70.06%** |
| AWQ-W3-G32 | 3.59 | 13.49 | 84.84% | **50.80%** | 54.10% | **86.15%** | 71.69% | 69.96% |
| BPDQ-W3-G64 | 4.00 | 13.29 | **85.67%** | 47.60% | 54.78% | 85.93% | 72.23% | 69.95% |
| GPTQ-W3-G64 | 3.30 | **13.50** | 79.45% | 46.20% | 48.12% | 85.66% | **72.58%** | 68.54% |
| AWQ-W3-G64 | 3.30 | 13.75 | 83.40% | 45.20% | 52.56% | 85.54% | 71.65% | 68.84% |
| BPDQ-W3-G128 | 3.50 | 13.71 | **83.85%** | **47.80%** | **55.80%** | **86.09%** | 71.51% | **70.01%** |
| GPTQ-W2-G32 | 2.56 | 22.05 | 0.38% | 2.60% | 30.80% | 63.98% | 57.18% | 32.47% |
| AWQ-W2-G32 | 2.56 | 2.4E+7 | 1.44% | 3.40% | 34.47% | 65.90% | 44.90% | 46.07% |
| BPDQ-W2-G64 | 2.75 | **18.83** | **52.99%** | **14.80%** | **44.37%** | **84.31%** | **62.15%** | **58.70%** |
| GPTQ-W2-G64 | 2.28 | 30.30 | 0.00% | 1.40% | 27.05% | 52.51% | 49.91% | 25.35% |
| AWQ-W2-G64 | 2.28 | 8.9E+10 | 0.00% | 0.00% | 27.30% | 61.68% | 27.92% | 23.05% |
| BPDQ-W2-G128 | 2.38 | **20.46** | **40.79%** | **10.20%** | **42.58%** | **80.83%** | **61.34%** | **55.13%** |
| *Qwen3-14B* | *16* | *10.78* | *88.02%* | *53.00%* | *60.32%* | *89.30%* | *78.82%* | *77.14%* |
| GPTQ-W4-G64 | 4.31 | **10.98** | **89.08%** | 52.80% | 61.09% | 88.87% | **78.52%** | **76.51%** |
| AWQ-W4-G64 | 4.31 | 11.29 | 88.02% | 52.00% | **61.26%** | **89.36%** | 78.32% | 76.02% |
| BPDQ-W4-G128 | 4.63 | 11.05 | 88.17% | **54.20%** | 60.24% | 89.24% | 78.33% | 75.99% |
| GPTQ-W3-G32 | 3.59 | **11.37** | 84.46% | 48.40% | 59.56% | 87.52% | **77.69%** | 74.48% |
| AWQ-W3-G32 | 3.59 | 11.86 | 87.64% | 50.80% | 59.04% | **88.87%** | 76.95% | **75.15%** |
| BPDQ-W3-G64 | 4.00 | 11.51 | **89.16%** | **52.80%** | **60.84%** | 88.78% | 76.91% | 74.22% |
| GPTQ-W3-G64 | 3.30 | **11.64** | 87.26% | 49.80% | **59.81%** | 87.89% | **77.04%** | 74.16% |
| AWQ-W3-G64 | 3.30 | 12.32 | 88.78% | 50.20% | 56.66% | 87.86% | 76.09% | 73.27% |
| BPDQ-W3-G128 | 3.50 | 11.84 | **88.86%** | **52.20%** | 58.62% | **89.08%** | 76.61% | **74.95%** |
| GPTQ-W2-G32 | 2.56 | 16.31 | 23.81% | 8.20% | 35.49% | 72.20% | 66.23% | 50.84% |
| AWQ-W2-G32 | 2.56 | 3.1E+6 | 0.53% | 0.00% | 41.30% | 64.92% | 51.20% | 30.14% |
| BPDQ-W2-G64 | 2.75 | **15.32** | **71.80%** | **34.80%** | **51.11%** | **87.40%** | **69.01%** | **66.08%** |
| GPTQ-W2-G64 | 2.28 | 20.09 | 4.09% | 3.80% | 29.78% | 64.40% | 60.28% | 29.90% |
| AWQ-W2-G64 | 2.28 | 4.8E+8 | 0.00% | 0.00% | 26.96% | 59.94% | 27.20% | 26.21% |
| BPDQ-W2-G128 | 2.38 | **16.69** | **59.74%** | **25.40%** | **50.43%** | **87.58%** | **67.86%** | **64.02%** |

*Table 6.* Evaluation results on Llama3.1-8B, Gemma2-9B, and Phi4-14B

| Model | Wiki2 ↓ | GSM8K ↑ | MATH500 ↑ | ARC-C ↑ | BoolQ ↑ | HellaS ↑ | MMLU ↑ | Cost (min) ↓ |
|---|---|---|---|---|---|---|---|---|
| *Llama3.1-8B* | *8.83* | *70.36%* | *36.20%* | *55.38%* | *85.41%* | *79.55%* | *68.43%* | – |
| GPTQ-W4-G32 | **9.00** | 65.88% | **33.60%** | **54.61%** | 84.65% | 79.08% | **67.53%** | **11** |
| AnyBCQ-W4-G64 | 9.27 | 65.73% | 31.00% | 54.01% | **85.14%** | 78.69% | 66.56% | 178 |
| BPDQ-W4-G64 | 9.19 | **74.00%** | 33.20% | 54.44% | 84.80% | **79.25%** | 66.98% | 19 |
| GPTQ-W3-G32 | **9.94** | 61.03% | 24.20% | 49.66% | 83.70% | **77.74%** | **63.25%** | **11** |
| AnyBCQ-W3-G64 | 10.16 | 61.61% | 24.60% | 50.38% | 83.97% | 77.16% | 62.76% | 173 |
| BPDQ-W3-G64 | 10.30 | **64.44%** | **24.80%** | 52.73% | **84.59%** | 77.28% | 62.80% | 18 |
| GPTQ-W2-G32 | 24.60 | 0.61% | 1.40% | 34.30% | 61.96% | 60.79% | 28.99% | **10** |
| AnyBCQ-W2-G64 | **18.70** | 4.93% | 2.60% | 38.74% | 75.08% | **65.60%** | 42.14% | 170 |
| BPDQ-W2-G64 | 21.70 | **15.92%** | **3.40%** | 42.58% | 78.53% | 64.46% | **48.00%** | 18 |
| *Gemma2-9B* | *10.54* | *68.08%* | *32.20%* | *65.78%* | *84.16%* | *80.00%* | *68.96%* | – |
| GPTQ-W4-G32 | 10.76 | **68.39%** | 32.40% | **65.44%** | 84.68% | 79.60% | **68.05%** | **15** |
| AnyBCQ-W4-G64 | 11.10 | 63.71% | 31.40% | 65.05% | **84.82%** | 79.56% | 67.14% | 221 |
| BPDQ-W4-G64 | **10.75** | 66.26% | **33.00%** | 65.10% | 84.19% | **79.82%** | 67.89% | 24 |
| GPTQ-W3-G32 | **11.47** | 60.58% | 27.00% | **64.25%** | 83.91% | **78.80%** | **66.03%** | **15** |
| AnyBCQ-W3-G64 | 11.84 | 58.77% | 28.80% | 62.12% | 83.87% | 78.01% | 65.54% | 216 |
| BPDQ-W3-G64 | 11.70 | **60.96%** | **29.60%** | 62.29% | **84.68%** | 78.17% | 65.60% | 24 |
| GPTQ-W2-G32 | 20.33 | 5.08% | 4.00% | 44.88% | 76.48% | **70.37%** | 42.80% | **15** |
| AnyBCQ-W2-G64 | 17.74 | 10.52% | 6.20% | 46.89% | 76.78% | 70.13% | 47.49% | 211 |
| BPDQ-W2-G64 | **17.68** | **25.63%** | **7.40%** | 47.61% | **77.68%** | 70.24% | **50.75%** | 24 |
| *Phi4-14B* | *7.76* | *89.76%* | *48.80%* | *55.55%* | *86.09%* | *81.98%* | *76.89%* | – |
| GPTQ-W4-G32 | 7.86 | **90.45%** | **47.80%** | 55.72% | 86.06% | **81.72%** | 76.71% | **15** |
| AnyBCQ-W4-G64 | 7.93 | 89.80% | 46.60% | 56.03% | **86.39%** | 80.57% | 77.15% | 348 |
| BPDQ-W4-G64 | **7.84** | 90.07% | **47.80%** | 57.08% | 85.81% | 81.46% | **77.49%** | 24 |
| GPTQ-W3-G32 | 8.22 | 87.57% | 43.40% | 54.52% | **86.12%** | 81.16% | 75.35% | **16** |
| AnyBCQ-W3-G64 | 8.56 | 86.76% | 44.00% | 56.58% | 85.96% | 80.94% | 76.10% | 339 |
| BPDQ-W3-G64 | **8.15** | **87.64%** | **45.20%** | 57.42% | 86.09% | **81.22%** | 76.24% | 24 |
| GPTQ-W2-G32 | 18.66 | 5.99% | 2.80% | 41.04% | 63.55% | 66.68% | 38.44% | **15** |
| AnyBCQ-W2-G64 | 12.28 | 61.57% | 22.80% | 50.36% | 83.22% | 72.21% | 64.96% | 331 |
| BPDQ-W2-G64 | **11.89** | **66.19%** | **23.60%** | 50.51% | **84.07%** | 72.74% | **66.61%** | 24 |

*Table 7.* Comparison with additional bit-plane and VQ-based methods on LLaMA2-7B.

| Model | Wiki2 ↓ | GSM8K ↑ | MATH500 ↑ | ARC-C ↑ | BoolQ ↑ | HellaS ↑ | MMLU ↑ |
|---|---|---|---|---|---|---|---|
| *LLaMA2-7B* | *8.75* | *13.50%* | *3.80%* | *44.71%* | *79.36%* | *76.20%* | *40.93%* |
| GPTQ-W3-G32 | **9.54** | 7.87% | 2.20% | **42.97%** | 77.22% | **73.92%** | 38.21% |
| Any-Precision LLM-W3-G64 | 12.07 | 0.64% | 0.00% | 39.76% | 75.11% | 71.73% | 36.38% |
| ShiftAddLLM-W3-G64 | 11.93 | 1.53% | 0.20% | 40.63% | 75.46% | 72.34% | 36.45% |
| AnyBCQ-W3-G64 | 11.25 | 2.61% | 1.00% | 40.92% | 76.57% | 72.85% | 36.98% |
| AQLM-W3 | 9.60 | 8.07% | 2.00% | 41.04% | 77.85% | 73.16% | 38.52% |
| QuIP#-W3 | 9.63 | 8.22% | **2.80%** | 41.56% | 78.20% | 73.34% | 39.07% |
| VPTQ-W3 | 9.65 | **8.36%** | **2.80%** | 41.73% | 78.18% | 73.71% | 39.20% |
| BPDQ-W3-G64 | 9.60 | 8.19% | 2.40% | 42.41% | **78.26%** | 73.68% | **39.30%** |
| GPTQ-W2-G32 | 19.18 | 0.91% | 1.60% | 32.39% | 59.63% | 61.14% | 26.33% |
| Any-Precision LLM-W2-G64 | 18.88 | 0.00% | 0.00% | 22.48% | 61.74% | 30.27% | 25.11% |
| ShiftAddLLM-W2-G64 | 18.34 | 0.00% | 0.00% | 24.75% | 63.10% | 43.98% | 25.38% |
| AnyBCQ-W2-G64 | 18.19 | 0.00% | 0.00% | 31.83% | 63.08% | 61.33% | 28.84% |
| AQLM-W2 | 17.07 | 0.95% | 1.40% | 32.86% | 71.09% | 61.25% | 29.87% |
| QuIP#-W2 | 17.05 | 1.26% | 1.80% | 33.51% | **71.68%** | 61.48% | 30.06% |
| VPTQ-W2 | **17.02** | **1.63%** | **2.00%** | 34.15% | 71.58% | **61.73%** | **30.25%** |
| BPDQ-W2-G64 | 17.05 | 1.30% | **2.00%** | 33.70% | 70.95% | 61.59% | 29.32% |

