# OpenReview forum: "BPDQ: Bit-Plane Decomposition Quantization on a Variable Grid for Large Language Models"
_ICML.cc/2026/Conference — ICML 2026 regular_

### Official Review · Reviewer_qvHN · 2026-02-15

**Soundness:** 3
**Presentation:** 3
**Significance:** 3
**Originality:** 3
**Overall Recommendation:** 4
**Confidence:** 4

**Summary:**

This paper proposes BPDQ,  a new PTQ algorithm that targets to alleviate the accuracy degradation observed in the 2-3 bit regime.

BPDQ builds on two established paradigms: the Hessian-based optimization objective utilized in GPTQ and the bit-plane decomposition structure recently applied in AnyBCQ. Specifically, similarly to GPTQ, BPDQ relies on calibration data to construct an approximation of the Hessian and use it to minimize the error at the layer's output. However, unlike in GPTQ, BPDQ uses bit-plane decomposition to construct a per-group-optimized non-uniform grid.
Intuitively, this construction provides a new operating point for GPTQ-like methods between the uniform grid used by GPTQ and the overhead of using a full codebook.

The procedure alternates between two distinct phases: a step where scalar coefficients are fixed to determine the optimal bit assignments via an exhaustive local search over the $2^b$ possible levels, and a step where these bit assignments are frozen to refit the scalar coefficients using a least-squares solver. Importantly, the method introduces a "Delta Correction" term to synchronize the error-propagation state during these updates. This ensures that the iterative refinement of the local quantization grid remains consistent with the greedy layer's error-compensation mechanism inherent to GPTQ-like frameworks.

In terms of implementation, for a target of $b$ bits per parameter, BPDQ utilizes $b+1$ half-precision scalar coefficients per group. Although this incurs higher metadata overhead than standard scale-and-zero formulations, the increased expressivity allows the method to use larger group sizes while maintaining good fidelity.

The theoretical analysis provided in the appendices shows that, considering the same group size, this variable grid strictly subsumes standard uniform quantization.

Empirically, BPDQ outperforms baselines such as GPTQ and AWQ across 2-bit and 3-bit settings on various models and benchmarks. The authors intend to release their source code and custom kernels.

**Compliance With Llm Reviewing Policy:**

Affirmed.

**Final Justification:**

The rebuttal did not address (W1-W4). Nevertheless, I see merit in this work in its current form and therefore decided to keep my positive scope.

**Key Questions For Authors:**

Q1: How does BPDQ compare to quantization methods like [1] and [2]?

Q2: The theoretical analysis proves that BPDQ is superior to uniform quantization for a fixed group size $g$. However, given that BPDQ requires $b+1$ metadata scalars, one must practically increase $g$ (e.g., to 256) to match the effective bitrate of baselines like GPTQ (which use $g=64$). Can you provide a theoretical bound on the accuracy loss due to this coarser granularity?

Q3: The paper argues that Vector Quantization (VQ) is computationally expensive and "slower." Is this referring to the offline calibration cost or the online inference latency? If inference speeds are comparable on modern hardware, why should one prefer BPDQ over VQ?

Q4: Have you evaluated BPDQ on Mixture-of-Experts (MoE) architectures (e.g., Mixtral, DeepSeek-MoE)? Given that MoEs exhibit sparsity and varying expert utilization, it is unclear how well BPDQ generalizes to this setting.

Q5: How does the proposed bit-plane kernel compare to native hardware support for emerging micro-scaling formats (e.g., MXFP4)? Specifically, does the bit-wise summation introduce instruction-bound latency that might prevent the method from scaling to future hardware generations compared to native FP4/INT4 execution?

**Limitations:**

The paper discusses limitations in Section 6. It is desired to add or discuss the following possible limitations as well:

- Usage of calibration data. During inference, out-of-distribution data may cause collapse.
- Inference speed. Evaluation of inference-optimized frameworks (e.g., vLLM) and comparisons with emerging HW-supported micro-scaling formats are missing.

**Strengths And Weaknesses:**

Strengths:

(S1) The paper presents a clever combination of two known paradigms: the Hessian-based "Optimal Brain Compression" objective (from GPTQ) and the flexible data structure of Bit-Plane Decomposition (used in AnyBCQ).

(S2) The method is supported by theoretical analysis. The derivation of the "Delta Correction" term is a nice contribution, ensuring that the iterative greedy local optimization procedure remains consistent with the output-minimization objective.

(S3) The evaluation covers different models and benchmarks. The results demonstrate improved performance in the 2-3 bit settings.


Weaknesses:

(W1) The context of the contribution is slightly obscured by the omission of relevant related work. Specifically, the paper overlooks:

 - Calibration-free methods that optimize quantization values per group (e.g., [1]).

 - Hierarchical quantization schemes commonly used in deployment, such as the GGUF/K-quants format (e.g., [2]).

Discussing or comparing against these baselines is desired to better position the contribution in the scope of PTQ.


(W2) There is a discrepancy between the theoretical analysis and the experimental implementation. The theoretical proofs assume a direct comparison of grids under identical conditions. However, to maintain a competitive bit budget, the method requires larger group sizes ($g=256$) than the baselines ($g=64/128$). The analysis does not account for the potential fidelity loss associated with this coarser granularity, making the "strict superiority" claim less clear in practice.

(W3) The reported inference latency shows no advantage over BF16. The paper does not discuss how this software-based bit-plane arithmetic compares with emerging hardware-native formats such as MXFP4 or NVFP4. Moreover, evaluating within an optimized framework (e.g., vLLM) is desired to reveal actual bottlenecks (memory bandwidth vs. instruction overhead).

(W4) The paper argues that VQ is slower, but this conflates quantization time (offline) with inference time (online). If BPDQ's inference relies on summing bit-planes and VQ relies on table lookups, the latency difference might be small. If the slowness refers only to the one-time offline quantization process, it is not clear why this is a significant drawback for a PTQ method intended for mass distribution.

(W5) The evaluation is restricted to dense models (Llama/Qwen) and does not consider Mixture-of-Experts architectures (e.g., Mixtral, DeepSeek-MoE). MoE models present unique quantization challenges due to their sparse activation patterns and variation in expert utilization. Given that MoEs are currently dominant in high-performance inference, demonstrating that BPDQ’s variable grid and Hessian-based updates remain robust across diverse experts is desired for broad applicability.



[1] Ben-Basat, Ran, et al. "Optimal and approximate adaptive stochastic quantization." Advances in Neural Information Processing Systems 37 (2024): 94265-94291.

[2] Gerganov, G. (2023). Llama.cpp: LLM inference in C/C++. GitHub repository. https://github.com/ggerganov/llama.cpp

---

> ### Author Rebuttal · Authors · 2026-03-30
>
> We sincerely thank the reviewer for recognizing the value of our **Delta Correction**, **theoretical analysis**, and **comprehensive evaluation (9 models, >8 tasks)**.
> To clarify a factual distinction: BPDQ is the **first** to introduce Bit-Plane Decomposition (BPD) to LLM quantization. AnyBCQ utilizes progressive residual quantization, not BPD.
>
> > W1/Q1: baselines
>
> > A1: We appreciate you broadening our perspective with these references. BPDQ is a deterministic weight-only PTQ method, whereas QUIVER [1] addresses the Adaptive Stochastic Quantization (ASQ) problem. In the quantization of LLMs, QUIVER is more suited for on-the-fly activation quantization. Meanwhile, K-quants (GGUF) [2] is a non-uniform/mixed-bit format and engineering implementation optimized for CPU SIMD instructions. Due to its complex memory layout (e.g., super-/sub-block ), it is unsuited to GPU Tensor Cores and cannot accelerate matrix multiplication. In contrast, BPDQ's bit-plane format offers broad adaptability across GPUs, CPUs, ASICs, and FPGAs.
> >
> > Furthermore, we also evaluated bit-plane (Any-Precision LLM, ShiftAddLLM) and VQ methods (AQLM, QuIP#).  BPDQ is highly competitive in accuracy while providing much lower quantization costs and a hardware-friendly uniform format.
> >
> > Please see "**Positioning BPDQ (SQ) vs. VPTQ (VQ)**" in our response to Reviewer rcAB for more details.
>
> **Table: Comparison with additional bit-plane and VQ methods on LLaMA2-7B** in our response to Reviewer Z3hJ.
>
> ---
>
> > W2/Q2: group sizes
>
> > A2: We clarify that the **variable grid's superiority** holds even under coarser granularity:
> >
> > (1) As noted in Appendix (Remark: Geometric Degrees of Freedom), a fixed grid enforces global rigidity. Regardless of group size, all groups in GPTQ share the exact same shape-invariant template. In contrast, BPDQ's variable grid introduces immense adaptability tailored for each individual group (millions to billions of groups per model), which unlocks significant representational capacity.
> >
> > (2) Despite larger groups, BPDQ-W2-G256 (2.19 BPW) outperforms GPTQ-W2-G64/32 (2.28/2.56 BPW) on Qwen2.5-72B (Table 1) This confirms that for massive models, the variable grid's representational gain outweighs the impact of coarser granularity. (Results excerpted in Reviewer hxps A2)
>
> ---
>
> > W3/Q5: MXFP4, NVFP4, vLLM
>
> > A3: While bit-plane formats currently lack the native Tensor Core support of MXFP4, our LUT-GEMM kernel demonstrates its efficiency feasibility. Designing custom CPU/GPU kernels and integrating BPDQ into optimized frameworks like vLLM requires substantial engineering effort, which is a key direction for our future work. For detailed discussions on bitwise efficiency, please refer to [Lut-gemm,Flightllm,Dfx].
>
> ---
>
> > W4/Q3: VQ
>
> > A4: (1) In our paper: "VQ methods suffer from prohibitive computational costs during codebook optimization" and we explicitly distinguished offline cost (min) and online latency (ms) in Table 3.
> >
> > (2) Regarding offline cost: While PTQ is a one-time process, the rapid iteration and massive scale of modern LLMs (e.g., Mistral-Large-3-675B) make it critical. Quantizing a 72B model takes ~4.3H with BPDQ vs ~76.4H with VPTQ. This >15× offline speedup is a highly significant practical advantage for resource-constrained deployments.
> >
> > (3) Regarding inference: Beyond offline speed, BPDQ's weight format (bit-planes) is uniform and suitable for diverse hardware deployments (e.g., FPGA, ASIC, NPU). Conversely, VPTQ relies on discrete indices and lookup tables (LUTs). While efficient on NVIDIA GPUs utilizing SRAM, this indirect memory access and fragmented data structure break the continuous dataflow required by edge NPUs, significantly limiting its general deployability.
>
> ---
>
> > W5/Q4: MoE
>
> > A5: BPDQ seamlessly extends to MoE models. Evaluation on Qwen2.5-35B-A3B shows BPDQ maintains high 2-bit fidelity:
>
> | Model | Wiki2 | MATH500 | ARC-C | HellaS |
> | :--- | :--- | :--- | :--- | :--- |
> | Qwen3.5-35B-A3B | 7.46 | 54.60% | 61.77% | 82.37% |
> | GPTQ-W2-G64 | 13.45 | 6.60% | 47.44% | 71.47% |
> | BPDQ-W2-G128 | 10.97 | 33.00% | 59.47% | 75.06% |
>
> ---
>
> > Limitations
>
> > A6: (1) We acknowledge the reliance on calibration data, which is standard practice for PTQ methods (e.g., GPTQ, AWQ). However, across our evaluation (8 tasks), we did not observe OOD collapse. While task-specific calibration can boost specific metrics, we argue that the severe accuracy collapse at ultra-low bits is primarily limited by the "fixed-grid" restriction, rather than OOD data. Moreover, calibration-free methods are indeed a valuable direction for future exploration.
> >
> > (2) Further integration with vLLM and exploring hardware-level efficiency are important. We appreciate you pointing this out, as this is exactly the direction of our ongoing efforts.
>
> Thank you again for your insightful and positive review. If our response has effectively addressed your concerns, we'd be grateful if you consider further improving your score.

---

> > ### Author Rebuttal · Reviewer_qvHN · 2026-04-03
> >
> > Thank you for the detailed rebuttal. I appreciate the additional experiment that partially addresses (W5).
> >
> > However, (W1)-(W4) remain unsolved.
> >
> > Therefore, I will keep my score.

---

> > > ### Author Response · Authors · 2026-04-05
> > >
> > > Thank you for your response and for acknowledging our additional experiment extending BPDQ to **MoE** architectures.
> > >
> > > Finally, we want to reiterate that we are the first to identify the "fixed grid" constraint as the fundamental limitation in ultra-low-bit PTQ and solve it backed by rigorous Hessian-based derivation.  As summarized in our contributions:
> > > *   **Insight:** We identify the shape invariance of fixed quantization grids as the fundamental constraint on optimization-based PTQ in low-bit regimes.
> > > *   **Methodology:** We formulate a rigorous optimization framework that extends optimization-based PTQ to variable grids.
> > > *   **Performance:** Extensive experiments across language understanding, reasoning, and long-context tasks demonstrate BPDQ's consistently high fidelity in low-bit regimes. Furthermore, we confirm that BPDQ inherently preserves critical outliers through activation analysis.
> > >
> > > Throughout our experiments, the comparison between GPTQ (fixed grid) and BPDQ (variable grid) serves as the most comprehensive ablation study. Both share the same Hessian framework. Breaking the fixed-grid limitation to rescue the 2-bit accuracy collapse is exactly what we want to present to the community (without mixed-precision, extra inference structures or fine-tuning).
> > >
> > > ---
> > >
> > > **Global Updates During Rebuttal:**
> > >
> > > We respectfully invite you to review the new experiments and insights we provided to other reviewers during this rebuttal phase.
> > >
> > > *   **Comprehensive AnyBCQ Baselines:** As detailed in our response to Reviewer Z3hJ, we evaluated BPDQ on AnyBCQ's models (Llama3.1, Gemma2, and Phi4). BPDQ consistently achieves better 2-bit accuracy. Notably, AnyBCQ takes 9 to 14 times longer to quantize than BPDQ.
> > >
> > > *   **Clarification on SQ vs. VQ:** Reviewer Z3hJ acknowledged that our "Positioning BPDQ (SQ) vs. VPTQ (VQ)" discussion (detailed in our response to rcAB) is helpful. This clarifies that while VQ methods pursue fidelity, BPDQ (SQ) provides a highly competitive alternative with a fraction of the quantization cost and uniform hardware adaptability.
> > >
> > > *   **BPW Ablation:** We demonstrated that BPDQ-W2-G256 (2.19 BPW) significantly outperforms GPTQ-W2-G32 (2.56 BPW) on Qwen2.5-72B. This proves that breaking the fixed-grid limitation fundamentally rescues the 2-bit collapse, rather than relying on a higher BPW parameter allocation.

---

### Official Review · Reviewer_hxps · 2026-03-10

**Soundness:** 2
**Presentation:** 3
**Significance:** 2
**Originality:** 2
**Overall Recommendation:** 3
**Confidence:** 5

**Summary:**

This paper proposes Bit-Plane Decomposition Quantization (BPDQ), a Post-Training Quantization (PTQ) method aimed at improving the performance of Large Language Models (LLMs) in the 2-3 bit regime. The authors argue that the performance collapse of existing optimal-PTQ methods (like GPTQ) at ultra-low bit-widths is caused by the rigidity of fixed quantization grids. To address this, BPDQ constructs a "variable grid" using bit-planes and scalar coefficients. The method iteratively refines these bit-planes and coefficients while applying a delta correction to maintain error-propagation consistency within the Hessian-induced geometry.

**Compliance With Llm Reviewing Policy:**

Affirmed.

**Final Justification:**

The additional evaluation data provided in the rebuttal seems questionable. For instance, the LLaMA2-7B result on GSM8K is notably lower than my independent testing results. Thus, I am keeping my score unchanged.

**Key Questions For Authors:**

1. Comparison with SOTA: Can the authors provide a comprehensive comparison against QuIP# \ QTIP \ AQLM \ EfficientQAT on the LLaMA or Qwen models?

2. To prove the algorithmic superiority of the variable grid, could the authors provide an ablation study comparing BPDQ and GPTQ under the same bits-per-weight (BPW)? Or with same/higher accuracy at lower BPW?

**Strengths And Weaknesses:**

Strengths:
1. The insight that the shape-invariant nature of fixed grids restricts the feasible solution set for Hessian-based error minimization is well-founded.
2. The proposed alternating optimization combined with the Delta Correction mechanism ensures mathematical consistency with the optimal-PTQ objective.
3. By decomposing weights into bit-planes, the method naturally supports efficient bit-parallel arithmetic and LUT-based inference, which is practical for deployment.

Weaknesses:
1. Missing Critical State-of-the-Art Baselines: The evaluation heavily relies on demonstrating superiority over GPTQ and AWQ at 2-bit; beating them is not a sufficient benchmark for a 2025 or 2026 submission. The authors omit comparisons with the actual state-of-the-art 2-bit PTQ methods, most notably the QuIP family (QuIP# and QTIP), AQLM and EfficientQAT.
2. In Table 1, the authors compare BPDQ-W2-G64 (2.75 BPW) against GPTQ-W2-G32 (2.56 BPW). This makes it difficult to isolate whether the performance gain comes from the proposed algorithmic design or simply the different parameter allocation strategy.

---

> ### Author Rebuttal · Authors · 2026-03-30
>
> We thank the reviewer for recognizing our **well-founded insight into the "fixed-grid" restriction**, the **mathematical consistency**  and the **practical efficiency** of BPDQ.
> We will address your questions below to further clarify the practical value of BPDQ.
>
> > W1: Missing Critical State-of-the-Art Baselines: ...
> >
> > Q1: Comparison with SOTA: ...
>
> > A1: To fully address your concern, we have evaluated bit-plane methods (Any-Precision LLM and ShiftAddLLM) and VQ methods (AQLM and QuIP#) on LLaMA2-7B. As shown in the table below, BPDQ maintains strong competitiveness in accuracy while offering significantly lower quantization costs and a uniform, hardware-friendly data format.
> >
> > For a detailed BPDQ vs. VQ discussion, please see "**Positioning BPDQ (SQ) vs. VPTQ (VQ)**" in our response to Reviewer rcAB.
>
> **Table: Comparison with additional bit-plane and VQ methods on LLaMA2-7B.**
>
> | Model | Wiki2 | GSM8K | MATH500 | ARC-C | BoolQ | HellaS | MMLU |
> | :--- | :--- | :--- | :--- | :--- | :--- | :--- | :--- |
> | Llama2-7B | 8.75 | 13.50% | 3.80% | 44.71% | 79.36% | 76.20% | 40.93% |
> |**3-Bit**||||||||
> | GPTQ-W3-G32 | **9.54** | 7.87% | 2.20% | **42.97%** | 77.22% | **73.92%** | 38.21% |
> | Any-Precision LLM-W3-G64 | 12.07 | 0.64% | 0% | 39.76% | 75.11% | 71.73% | 36.38% |
> | ShiftAddLLM-W3-G64 | 11.93 | 1.53% | 0.20% | 40.63% | 75.46% | 72.34% | 36.45% |
> | AnyBCQ-W3-G64 | 11.25 | 2.61% | 1.00% | 40.92% | 76.57% | 72.85% | 36.98% |
> | AQLM-W3 | $\underline{9.60}$ | 8.07% | 2.00% | 41.04% | 77.85% | 73.16% | 38.52% |
> | QuIP#-W3 | 9.63 | $\underline{8.22}$% | **2.80%** | 41.56% | $\underline{78.20}$% | 73.34% | 39.07% |
> | VPTQ-W3 | 9.65 | **8.36%** | **2.80%** | 41.73% | 78.18% | $\underline{73.71}$% | $\underline{39.20}$% |
> | BPDQ-W3-G64 | $\underline{9.60}$ | 8.19% | $\underline{2.40}$% | $\underline{42.41}$% | **78.26%** | 73.68% | **39.30%** |
> |**2-Bit**||||||||
> | GPTQ-W2-G32 | 19.18 | 0.91% | 1.60% | 32.39% | 59.63% | 61.14% | 26.33% |
> | Any-Precision LLM-W2-G64 | 18.88 | 0% | 0% | 22.48% | 61.74% | 30.27% | 25.11% |
> | ShiftAddLLM-W2-G64 | 18.34 | 0% | 0% | 24.75% | 63.10% | 43.98% | 25.38% |
> | AnyBCQ-W2-G64 | 18.19 | 0% | 0% | 31.83% | 63.08% | 61.33% | 28.84% |
> | AQLM-W2 | 17.07 | 0.95% | 1.40% | 32.86% | 71.09% | 61.25% | 29.87% |
> | QuIP#-W2 | $\underline{17.05}$ | 1.26% | $\underline{1.80}$% | 33.51% | **71.68%** | 61.48% | $\underline{30.06}$% |
> | VPTQ-W2 | **17.02** | **1.63%** | **2.00%** | **34.15%** | $\underline{71.58}$% | **61.73%** | **30.25%** |
> | BPDQ-W2-G64 | $\underline{17.05}$ | $\underline{1.30}$% | **2.00%** | $\underline{33.70}$% | 70.95% | $\underline{61.59}$% | 29.32% |
>
> ---
>
>  > W2: In Table 1, the authors compare BPDQ-W2-G64 (2.75 BPW) against GPTQ-W2-G32 (2.56 BPW). ...
> >
> > Q2: To prove the algorithmic superiority of the variable grid, ...
>
> > A2: (1) Due to the inherent structural constraints of the bit-plane format, it is difficult to align the exact BPW with GPTQ. Therefore, BPDQ selects a larger group size (i.e., a coarser granularity) to approximate the BPW. Crucially, the actual model size (GB) difference is negligible in practice. Taking Qwen2.5-7B as an example:
> > - **W4**: GPTQ-W4-G64 (4.31BPW, 5.31GB) vs. BPDQ-W4-G128 (4.63BPW, 5.54GB)
> > - **W3**: GPTQ-W3-G32 (3.59BPW, 4.77GB) vs. BPDQ-W3-G64 (4.00BPW, 5.07GB)
> > - **W3**: GPTQ-W3-G64 (3.30BPW, 4.54GB) vs. BPDQ-W3-G128 (3.50BPW, 4.69GB)
> > - **W2**: GPTQ-W2-G32 (2.56BPW, 3.98GB) vs. BPDQ-W2-G64 (2.75BPW, 4.12GB)
> > - **W2**: GPTQ-W2-G64 (2.28BPW, 3.77GB) vs. BPDQ-W2-G128 (2.38BPW, 3.84GB)
> >
> > For the BPW gap (2.56 vs 2.75) you mentioned, the physical model size difference on a 7B model is merely **0.14 GB** (3.98 GB vs 4.12 GB). We believe this minimal footprint difference is acceptable.
> >
> > (2) Moreover, as shown in Table 1 for Qwen2.5-72B, BPDQ-W2-G256 (**BPW 2.19**) significantly outperforms both GPTQ-W2-G64 (**BPW 2.28**) and GPTQ-W2-G32 (**BPW 2.56**). This provides a clear example of achieving "higher accuracy at lower BPW". It fully demonstrates that breaking the fixed-grid limitation is crucial in the ultra-low-bit regime. For massive models like 72B, the variable grid effectively unlocks the representation potential and significantly boosts the performance of optimization-based PTQ methods.
>
> | Model | BPW | Wiki2 | GSM8K | MATH500 | ARC-C | BoolQ | HellaS | MMLU |
> | :--- | :--- | :--- | :--- | :--- | :--- | :--- | :--- | :--- |
> | GPTQ-W2-G32 | 2.56 | 10.01 | 63.46% | 28.40% | 53.16% | 86.21% | 78.60% | 69.59% |
> | GPTQ-W2-G64 | 2.28 | 12.47 | 40.49% | 14.40% | 41.89% | 79.79% | 74.69% | 62.18% |
> | BPDQ-W2-G256 | 2.19 | **8.94** | **83.85%** | **39.40%** | **60.24%** | **89.72%** | **81.69%** | **75.89%** |
>
> Thank you again for your valuable feedback. We hope our comprehensive comparisons and ablation analysis have effectively addressed your concerns regarding the baselines and BPW alignment. If so, we would greatly appreciate it if you could consider raising your score.

---

> > ### Author Rebuttal · Reviewer_hxps · 2026-04-04
> >
> > The results in Table seem quite unusual. The Wiki perplexity for LLaMA2-7B is higher than what has been reported in other papers. And the evaluation results on benchmarks like GSM8K and Math-500 are clearly problematic. Therefore, I will maintain my current score.

---

> > > ### Author Response · Authors · 2026-04-05
> > >
> > > Thank you for your acknowledgement.
> > >
> > > **1. Regarding the PPL:**
> > >
> > > We evaluated the models using `lm-eval v0.4.11` on the `EleutherAI___wikitext_document_level/wikitext-2-raw-v1` dataset. We will release the Llama2-7B BPDQ-W3/2-G64 and Qwen2.5-72B BPDQ-W2-G256 (BPW 2.19) checkpoints, and we welcome you to test them.
> > >
> > > **2. Regarding GSM8K and Math500:**
> > >
> > > The performance collapse of GPTQ/AWQ can be referenced in *"Evaluating Quantized Large Language Models"* and *"Quantization Hurts Reasoning? An Empirical Study on Quantized Reasoning Models"*.
> > >
> > > For how BPDQ addresses this problem, please refer to our theoretical explanations (Appendix A formalizes how this variable grid expands the feasible solution set, and Appendix B formalizes the consistency of BPDQ with Hessian-induced optimality), as well as the Activation Outlier Statistics in Section 4.3, which verify BPDQ's direct impact on preserving essential outliers.
> > >
> > > **3. Final Statement:**
> > >
> > > We want to reiterate that we are the first to identify the "fixed grid" constraint as the fundamental limitation in ultra-low-bit PTQ and solve it backed by rigorous Hessian-based derivation.  As summarized in our contributions:
> > > *   **Insight:** We identify the shape invariance of fixed quantization grids as the fundamental constraint on optimization-based PTQ in low-bit regimes.
> > > *   **Methodology:** We formulate a rigorous optimization framework that extends optimization-based PTQ to variable grids.
> > > *   **Performance:** Extensive experiments across language understanding, reasoning, and long-context tasks demonstrate BPDQ's consistently high fidelity in low-bit regimes. Furthermore, we confirm that BPDQ inherently preserves critical outliers through activation analysis.
> > >
> > > Throughout our experiments, the comparison between GPTQ (fixed grid) and BPDQ (variable grid) serves as the most comprehensive ablation study. Both share the same Hessian framework. Breaking the fixed-grid limitation to rescue the 2-bit accuracy collapse is exactly what we want to present to the community (without mixed-precision, extra inference structures or fine-tuning).
> > >
> > > ---
> > >
> > > **Global Updates During Rebuttal:**
> > >
> > > We respectfully invite you to review the new experiments and insights we provided to other reviewers during this rebuttal phase.
> > >
> > > *   **Comprehensive AnyBCQ Baselines:** As detailed in our response to Reviewer Z3hJ, we evaluated BPDQ on AnyBCQ's models (Llama3.1, Gemma2, and Phi4). BPDQ consistently achieves better 2-bit accuracy. Notably, AnyBCQ takes 9 to 14 times longer to quantize than BPDQ.
> > >
> > > *   **Clarification on SQ vs. VQ:** Reviewer Z3hJ acknowledged that our "Positioning BPDQ (SQ) vs. VPTQ (VQ)" discussion (detailed in our response to rcAB) is helpful. This clarifies that while VQ methods pursue fidelity, BPDQ (SQ) provides a highly competitive alternative with a fraction of the quantization cost and uniform hardware adaptability.
> > >
> > > *   **Extension to MoE Architectures:** Reviewer qvHN acknowledged our additional experiments extending BPDQ to MoE architectures. This proves that BPDQ is a generalizable framework capable of handling diverse model structures.

---

### Official Review · Reviewer_Z3hJ · 2026-03-11

**Soundness:** 3
**Presentation:** 3
**Significance:** 2
**Originality:** 3
**Overall Recommendation:** 4
**Confidence:** 3

**Summary:**

This paper proposes BPDQ designed to enable high-fidelity LLM inference at ultra-low bitwidths (2–3 bits). BPDQ introduces a variable quantization grid. By decomposing weights into bit-planes and group-wise scalar coefficients, it expands the feasible set for error minimization. Utilizing the Hessian-induced geometry from GPTQ, BPDQ iteratively refines these planes and coefficients, applying a "delta correction" to ensure optimization remains consistent with the output-aligned objective.

**Compliance With Llm Reviewing Policy:**

Affirmed.

**Final Justification:**

Thanks to the author's detailed responses. My concerns haven been mostly resolved. Thus, I would like to increase my score to 4.

**Key Questions For Authors:**

1. I found that the baseline quantization methods in this paper are quite old. SOTA uniform grid quantization methods methods like flatquant[1], BRQ[2] are not included for comparision. Is there any reason not comparing with these SOTA uniform grid quantization methods?


[1] Sun, Yuxuan, Ruikang Liu, Haoli Bai, Han Bao, Kang Zhao, Yuening Li, Xianzhi Yu et al. "FlatQuant: Flatness Matters for LLM Quantization." In Forty-second International Conference on Machine Learning.

[2] Shao, Yuantian, Peisong Wang, Yuanteng Chen, Chang Xu, Zhihui Wei, and Jian Cheng. "Block rotation is all you need for mxfp4 quantization." arXiv preprint arXiv:2511.04214 (2025).

**Limitations:**

yes

**Strengths And Weaknesses:**

Strengths:
1. This paper proposes a new framework targetting the hardware-friendly bit-plane quantization. The proposed method uses hessian-induced geometry with theoretical guarantees.
2. System efficiency profiling is provided for audience to understand the tradeoff and efficiency of the proposed method.

Weakness:

1. The evaluation models and baselines are not comprehensive enough. Being a bit-plane quantization method, the most important baseline is AnyBCQ. However, there is only one experiment (table 2) comparing proposed BPDQ method with AnyBCQ. It would be more convincing to add more experiements comparing with AnyBCQ, especially on models and benchmarks that was used in the original AnyBCQ paper.
2. In experiments (table 1 & 2), the improvement against baselines is rather marginal in some cases, but the bits-per-weight (BPW) is usually higher than baselines, suggesting BPDQ using more resorces. For example, In Table 1, for the Qwen2.5-72B model at 3-bit, BPDQ has a BPW of 3.50, whereas GPTQ and AWQ have a lower BPW of 3.30. At the same time, the PPL on wiki2 is only marginal better than GPTQ and AWQ (5.73 vs 6.04 and 5.85). This further weakens the emperical results.

---

> ### Author Rebuttal · Authors · 2026-03-30
>
> We thank the reviewer for recognizing **BPDQ's theoretical guarantees**, **hardware-friendly format**, and **system efficiency**. We address your questions below to further clarify its practical value.
>
> > W1: The evaluation ...
>
> > A1: We appreciate your constructive suggestion. To address your concern, we added **LLaMA2-7B** comparisons against **bit-plane (Any-Precision LLM, ShiftAddLLM, AnyBCQ)** and **VQ methods (AQLM, QuIP#, VPTQ)**. As shown below, BPDQ maintains strong competitiveness in accuracy while offering significantly lower quantization costs and a uniform, hardware-friendly data format.
> >
> > For a detailed BPDQ vs. VQ discussion, please see "**Positioning BPDQ (SQ) vs. VPTQ (VQ)**" in our response to Reviewer rcAB.
>
> **Table: Comparison with additional bit-plane and VQ methods on LLaMA2-7B.**
>
> | Model | Wiki2 | GSM8K | MATH500 | ARC-C | BoolQ | HellaS | MMLU |
> | :--- | :--- | :--- | :--- | :--- | :--- | :--- | :--- |
> | Llama2-7B | 8.75 | 13.50% | 3.80% | 44.71% | 79.36% | 76.20% | 40.93% |
> |**3-Bit**||||||||
> | GPTQ-W3-G32 | **9.54** | 7.87% | 2.20% | **42.97%** | 77.22% | **73.92%** | 38.21% |
> | Any-Precision LLM-W3-G64 | 12.07 | 0.64% | 0% | 39.76% | 75.11% | 71.73% | 36.38% |
> | ShiftAddLLM-W3-G64 | 11.93 | 1.53% | 0.20% | 40.63% | 75.46% | 72.34% | 36.45% |
> | AnyBCQ-W3-G64 | 11.25 | 2.61% | 1.00% | 40.92% | 76.57% | 72.85% | 36.98% |
> | AQLM-W3 | $\underline{9.60}$ | 8.07% | 2.00% | 41.04% | 77.85% | 73.16% | 38.52% |
> | QuIP#-W3 | 9.63 | $\underline{8.22}$% | **2.80%** | 41.56% | $\underline{78.20}$% | 73.34% | 39.07% |
> | VPTQ-W3 | 9.65 | **8.36%** | **2.80%** | 41.73% | 78.18% | $\underline{73.71}$% | $\underline{39.20}$% |
> | BPDQ-W3-G64 | $\underline{9.60}$ | 8.19% | $\underline{2.40}$% | $\underline{42.41}$% | **78.26%** | 73.68% | **39.30%** |
> |**2-Bit**||||||||
> | GPTQ-W2-G32 | 19.18 | 0.91% | 1.60% | 32.39% | 59.63% | 61.14% | 26.33% |
> | Any-Precision LLM-W2-G64 | 18.88 | 0% | 0% | 22.48% | 61.74% | 30.27% | 25.11% |
> | ShiftAddLLM-W2-G64 | 18.34 | 0% | 0% | 24.75% | 63.10% | 43.98% | 25.38% |
> | AnyBCQ-W2-G64 | 18.19 | 0% | 0% | 31.83% | 63.08% | 61.33% | 28.84% |
> | AQLM-W2 | 17.07 | 0.95% | 1.40% | 32.86% | 71.09% | 61.25% | 29.87% |
> | QuIP#-W2 | $\underline{17.05}$ | 1.26% | $\underline{1.80}$% | 33.51% | **71.68%** | 61.48% | $\underline{30.06}$% |
> | VPTQ-W2 | **17.02** | **1.63%** | **2.00%** | **34.15%** | $\underline{71.58}$% | **61.73%** | **30.25%** |
> | BPDQ-W2-G64 | $\underline{17.05}$ | $\underline{1.30}$% | **2.00%** | $\underline{33.70}$% | 70.95% | $\underline{61.59}$% | 29.32% |
>
> ---
>
>  > W2: In experiments (table 1 & 2), ...
>
> > A2: (1) **marginal**: At W4/W3, 16/8 selectable values provide sufficient resolution, rendering the "fixed-grid" constraint a secondary issue. This limitation becomes critical only at W2, where BPDQ’s variable grid prevents the fidelity collapse inherent in fixed grids. At W2, BPDQ significantly outperforms GPTQ/AWQ and performs on par with VPTQ, while offering lower quantization costs and superior hardware adaptability.
> >
> > (2) **BPW**: Due to the inherent structural constraints of the bit-plane format, it is difficult to align the exact BPW with GPTQ. Thus, BPDQ selects a larger group size (i.e., a coarser granularity) to approximate the BPW. Crucially, the actual model size (GB) difference is negligible in practice (e.g., Qwen2.5-7B):
> > - **W4**: GPTQ-W4-G64 (4.31BPW, 5.31GB) vs. BPDQ-W4-G128 (4.63BPW, 5.54GB)
> > - **W3**: GPTQ-W3-G32 (3.59BPW, 4.77GB) vs. BPDQ-W3-G64 (4.00BPW, 5.07GB)
> > - **W3**: GPTQ-W3-G64 (3.30BPW, 4.54GB) vs. BPDQ-W3-G128 (3.50BPW, 4.69GB)
> > - **W2**: GPTQ-W2-G32 (2.56BPW, 3.98GB) vs. BPDQ-W2-G64 (2.75BPW, 4.12GB)
> > - **W2**: GPTQ-W2-G64 (2.28BPW, 3.77GB) vs. BPDQ-W2-G128 (2.38BPW, 3.84GB)
> >
> > For the mentioned BPW gap (3.30 vs 3.50), the 7B model size difference is merely 0.15 GB (4.54 vs 4.69 GB), an acceptable minimal footprint overhead.
> >
> > Moreover, for Qwen2.5-72B, BPDQ-W2-G256 (BPW 2.19) significantly outperforms GPTQ-W2-G32 (BPW 2.56). This provides an example of achieving "higher accuracy at lower BPW". See Reviewer hxps A2(2) for details.
>
> ---
>
> > Q1: I found that the baseline ...
>
> > A3: For more comprehensive baseline comparisons, please refer to A1.
> > Regarding FlatQuant and BRQ:
> >
> > (1) **FlatQuant**: It requires introducing an affine transformation during inference to smooth outliers. It mainly focuses on W-A quantization. In contrast, BPDQ focuses on Uniform Grid Weight-only quantization without any extra operations or structural modifications during inference. Thus, they target different hardware deployment scenarios (e.g., bit-plane hardware efficiency on FPGA/ASIC).
> >
> > (2) **BRQ**: It is specifically designed for the `mxfp4` data type. Therefore, it was not considered suitable for our general W4/3/2 integer/bit-plane quantization comparisons.
>
> Thank you again for your constructive review. Hope our detailed response above can address your concerns and make you favorably raise your scores.

---

> > ### Author Rebuttal · Reviewer_Z3hJ · 2026-04-01
> >
> > Thank you for the detailed rebuttal and the additional experiments comparing BPDQ with bit-plane and VQ methods on LLaMA2-7B.
> >
> > I acknowledge the authors' responses to my concerns. The clarifications on BPW overhead (W2/W3) and the positioning of BPDQ relative to VQ methods are helpful. However, my primary concern regarding insufficient empirical support remains unresolved:
> >
> > 1. The additional AnyBCQ comparisons are limited to a single model (LLaMA2-7B) and do not extend to the broader set of models and benchmarks used in the original AnyBCQ paper, as I originally requested.
> >
> > 2. The marginal improvements at W3/W4 over strong baselines, which was a key concern, are not adequately addressed — the argument that fixed-grid constraints are only critical at W2 does not fully justify the overhead at higher bitwidths.
> >
> > I appreciate the authors' effort and the additional comparison against FlatQuant and BRQ. These clarify the scope of the work.
> >
> > Given that the empirical support remains limited, I will maintain my current rating of 3: Weak Reject.

---

> > > ### Author Response · Authors · 2026-04-05
> > >
> > > Thank you for your response and for acknowledging that the "**Positioning BPDQ (SQ) vs. VPTQ (VQ)**" is helpful.
> > >
> > > **1. Comparison with AnyBCQ's Models:**
> > >
> > > We evaluated the models from the AnyBCQ paper (Llama3.1, Gemma2, Phi4), and the results align with our claims. In the 2-bit regime, BPDQ consistently outperforms GPTQ and AnyBCQ, **especially on complex reasoning tasks like GSM8K & MATH500**. BPDQ also secures the most optimal results at 3-bit.
> > >
> > > |Model|Wiki2|GSM8K|MATH500|ARC-C|BoolQ|HellaS|MMLU|Cost(min)|
> > > |:---|:---|:---|:---|:---|:---|:---|:---|:---|
> > > |**Llama3.1-8B**|8.83|70.36%|36.20%|55.38%|85.41%|79.55%|68.43%|-|
> > > |GPTQ-W3-G32|**9.94**|61.03%|24.20%|49.66%|83.70%|**77.74%**|**63.25%**|11|
> > > |AnyBCQ-W3-G64|10.16|61.61%|24.60%|50.38%|83.97%|77.16%|62.76%|173|
> > > |**BPDQ-W3-G64**|10.30|**64.44%**|**24.80%**|**52.73%**|**84.59%**|77.28%|62.80%|18|
> > > |GPTQ-W2-G32|24.60|0.61%|1.40%|34.30%|61.96%|60.79%|28.99%|10|
> > > |AnyBCQ-W2-G64|**18.70**|4.93%|2.60%|38.74%|75.08%|**65.60%**|42.14%|170|
> > > |**BPDQ-W2-G64**|21.70|**15.92%**|**3.40%**|**42.58%**|**78.53%**|64.46%|**48.00%**|18|
> > > ||||||||||
> > > |**Gemma2-9B**|10.54|68.08%|32.20%|65.78%|84.16%|80.00%|68.96%|-|
> > > |GPTQ-W3-G32|**11.47**|60.58%|27.00%|**64.25%**|83.91%|**78.80%**|**66.03%**|15|
> > > |AnyBCQ-W3-G64|11.84|58.77%|28.80%|62.12%|83.87%|78.01%|65.54%|216|
> > > |**BPDQ-W3-G64**|11.70|**60.96%**|**29.60%**|62.29%|**84.68%**|78.17%|65.60%|24|
> > > |GPTQ-W2-G32|20.33|5.08%|4.00%|44.88%|76.48%|**70.37%**|42.80%|15|
> > > |AnyBCQ-W2-G64|17.74|10.52%|6.20%|46.89%|76.78%|70.13%|47.49%|211|
> > > |**BPDQ-W2-G64**|**17.68**|**25.63%**|**7.40%**|**47.61%**|**77.68%**|70.24%|**50.75%**|24|
> > > ||||||||||
> > > |**Phi4-14B**|7.76|89.76%|48.80%|55.55%|86.09%|81.98%|76.89%|-|
> > > |GPTQ-W3-G32|8.22|87.57%|43.40%|54.52%|**86.12%**|81.16%|75.35%|16|
> > > |AnyBCQ-W3-G64|8.56|86.76%|44.00%|56.58%|85.96%|80.94%|76.10%|339|
> > > |**BPDQ-W3-G64**|**8.15**|**87.64%**|**45.20%**|**57.42%**|86.09%|**81.22%**|**76.24%**|24|
> > > |GPTQ-W2-G32|18.66|5.99%|2.80%|41.04%|63.55%|66.68%|38.44%|15|
> > > |AnyBCQ-W2-G64|12.28|61.57%|22.80%|50.36%|83.22%|72.21%|64.96%|331|
> > > |**BPDQ-W2-G64**|**11.89**|**66.19%**|**23.60%**|**50.51%**|**84.07%**|**72.74%**|**66.61%**|24|
> > >
> > > Notably, AnyBCQ requires scale fine-tuning, making **its quantization time 9-14× slower than BPDQ**.
> > >
> > > *(Note: GPTQ/BPDQ time depends primarily on layer count, so Phi4-14B 40-layers and Gemma2-9B 42-layers take similar times).*
> > >
> > > **2. The "marginal" W4/W3 improvements:**
> > >
> > > As stated in our Abstract: *"While post-training quantization (PTQ) maintains high fidelity at 4-bit, it deteriorates at 2-3 bits."* Currently, 4-bit PTQ is mature. Because the 4-bit representation space is already sufficient, it is theoretically expected that improvements at W4/W3 are relatively marginal.
> > >
> > > Our core motivation strictly targets the ultra-low-bit limitation. As stated in our Abstract: *"Fundamentally, existing methods enforce a shape-invariant quantization grid for each group, severely restricting the feasible set for error minimization."*
> > >
> > > To prove this, as shown in the new table above, BPDQ consistently achieves the best results at 2-bit, especially on complex tasks like GSM8K & MATH500. Moreover, as shown in Table 1 for Qwen2.5-72B, BPDQ-W2-G256 (**BPW 2.19**) significantly outperforms GPTQ-W2-G32 (**BPW 2.56**). Even with a lower BPW, the variable grid effectively unlocks the representation potential and significantly boosts the performance for massive models.
> > >
> > > |Model|BPW|Wiki2|GSM8K|MATH500|ARC-C|BoolQ|HellaS|MMLU|
> > > |:---|:---|:---|:---|:---|:---|:---|:---|:---|
> > > | GPTQ-W2-G32 | 2.56 | 10.01 | 63.46% | 28.40% | 53.16% | 86.21% | 78.60% | 69.59% |
> > > | BPDQ-W2-G256 | 2.19 | **8.94** | **83.85%** | **39.40%** | **60.24%** | **89.72%** | **81.69%** | **75.89%** |
> > >
> > > *(We will release this Qwen2.5-72B BPDQ-W2-G256 checkpoint)*
> > >
> > > **3. Final Statement:**
> > >
> > > We want to reiterate that we are the first to identify the "fixed grid" constraint as the fundamental limitation in ultra-low-bit PTQ and solve it backed by rigorous Hessian-based derivation.  As summarized in our contributions:
> > > *   **Insight:** We identify the shape invariance of fixed quantization grids as the fundamental constraint on optimization-based PTQ in low-bit regimes.
> > > *   **Methodology:** We formulate a rigorous optimization framework that extends optimization-based PTQ to variable grids.
> > > *   **Performance:** Extensive experiments across language understanding, reasoning, and long-context tasks demonstrate BPDQ's consistently high fidelity in low-bit regimes. Furthermore, we confirm that BPDQ inherently preserves critical outliers through activation analysis.
> > >
> > > Throughout our experiments, the comparison between GPTQ (fixed grid) and BPDQ (variable grid) serves as the most comprehensive ablation study. Both share the same Hessian framework. Breaking the fixed-grid limitation to rescue the 2-bit accuracy collapse is exactly what we want to present to the community (without mixed-precision, extra inference structures or fine-tuning).

---

### Official Review · Reviewer_rcAB · 2026-03-13

**Soundness:** 3
**Presentation:** 2
**Significance:** 3
**Originality:** 3
**Overall Recommendation:** 4
**Confidence:** 5

**Summary:**

This paper argues that the failure of ultra-low-bit quantization mainly stems from the fixed grid quantizers. To address this, authors propose BPDQ which is a variable grid quantization method based on bit-plane decomposition. The method init quantization using uint8 and bit-plane decomposition, then iterately updates bit-planes and refit coefficients, and also introduce delta correction to maintain error-propagation consistency. BPDQ follows GPTQ-based quantizaiton framework. Experiments show that it achieves acc. improvements over GPTQ, AWQ, and AnyBCQ, although it does not surpass VPTQ.

**Compliance With Llm Reviewing Policy:**

Affirmed.

**Final Justification:**

The authors’ rebuttal addressed most of my main concerns and provided useful clarifications. While some minor issues remain, my overall assessment has improved. I therefore revise my recommendation to Weak Accept.

**Key Questions For Authors:**

- Can this method be extended to activation quantization?
- Why choose unit8 as initialization?
- in table2, VPTQ appears to be substantially better than BPDQ. However, Table 1 does not include VPTQ results. Could the authors provide a clearer comparison between BPDQ and VPTQ?
- in table3, the latency between VPTQ and BPDQ looks fairly similar, which does not seem to demonstrate a clear advantage for BPDQ. Could the authors more explicitly discuss the practical advantages and trade-offs of BPDQ?

**Limitations:**

yes

**Strengths And Weaknesses:**

### Strengths
1. the perspective that fixed grid quantization is the main bottleneck of ultra-low bit PTQ is meaningful
2. the procedure of optimizing variable grid is well-designed

### Weaknesses
1. improvements on W4/3 are marginal; W2 result is significantly bebind VPTQ.
2. ∼3× the quantization time of GPTQ on a single NVIDIA H20 GPU is not cheap
3. should include more UQ and VQ methods and compare acc and latency

---

> ### Author Rebuttal · Authors · 2026-03-30
>
> We thank the reviewer for recognizing the **significance of the "fixed-grid" bottleneck** and our **well-designed BPDQ procedure**.
>
> > **Positioning BPDQ (SQ) vs. VPTQ (VQ).**
> >
> > Regarding the concern that VPTQ surpasses BPDQ in W2 accuracy, we provide a comprehensive clarification:
> >
> >  $\quad$ BPDQ and VPTQ belong to two distinct quantization paradigms: Scalar Quantization (SQ) and Vector Quantization (VQ). As mentioned in our paper, "VQ methods suffer from prohibitive computational costs during codebook optimization." We included VPTQ because VQ methods serve as a top-performance reference for ultra-low bit quantization, rather than a direct baseline. While VPTQ achieves the highest accuracy, it incurs a prohibitive ~40× quantization time overhead relative to GPTQ (and other VQ methods like AQLM cost ~400×). In contrast, BPDQ only requires ~3× the quantization time of GPTQ.
> >
> >   $\quad$ Moreover, BPDQ's uniform weight format (bit-planes and scaling) is **hardware-agnostic**, making it exceptionally well-suited for edge deployments, including NPUs (e.g., Qualcomm, MTK, AMD), FPGAs, and ASICs. Conversely, VPTQ relies on discrete indices and lookup tables (LUTs). While efficient on NVIDIA GPUs utilizing SRAM, this indirect memory access and fragmented data structure break the continuous dataflow required by edge NPUs, significantly limiting its general deployability.
> >
> >   $\quad$ Consequently, the practical trade-off is clear: compared to VPTQ, BPDQ offers significantly lower quantization costs and superior hardware adaptability, while still maintaining high fidelity. (Detailed bit-plane hardware adaptability in [Lut-gemm, Flightllm, Dfx])
> >
> >   **NOTE**: BPDQ does not employ mixed precision, particular outlier handling, or fine-tuning. While these enhancements are entirely feasible and have been outlined in our Future Work, we intentionally presented the "cleanest" version of BPDQ. Our goal is to purely demonstrate how it breaks through the overlooked "fixed-grid" constraint in ultra-low bit quantization.
>
> Below, we address your concerns point by point.
>
> > W1: W4/3 are marginal ...
>
> > A1: At W4/W3, 16/8 selectable values provide sufficient resolution, making the "fixed-grid" constraint a secondary issue. The limitation only becomes critical at ultra low-bit (W2), where a rigid grid severely degrades fidelity. BPDQ’s variable grid breaks the bottleneck. Furthermore, our near-optimal results across all precisions demonstrate BPDQ's robustness as a general-purpose framework. (For W2 vs. VPTQ, see SQ vs. VQ above).
>
> ---
>
> > W2: ∼3× the quantization time
>
> > A2: We refactored BPDQ into GPTQModel v5.7.0 (algorithm unchanged). Tested on Qwen2.5-7B (1×A6000), 1-iteration BPDQ matches GPTQ's cost while yielding substantially higher accuracy (e.g., GSM8K: 13.80% vs 3.00%, MMLU: 57.96% vs 37.12%). Moreover, this refactoring drops the overhead for 5 iterations from ~3× to ~2×. This ~2-3× cost is practical, especially compared to VPTQ's ~40× overhead.
>
> **Table: Iterations and Time Cost** in https://anonymous.4open.science/r/START-02D2
>
> ---
>
> > W3: should include more UQ and VQ
>
> > A3: VPTQ is a representative of UQ/VQ methods. To fully address your concern, we have evaluated bit-plane methods (Any-Precision LLM and ShiftAddLLM) and other UQ/VQ methods (AQLM and QuIP#) on LLaMA2-7B. As shown, BPDQ maintains strong competitiveness in accuracy while offering significantly lower quantization costs and a uniform, hardware-friendly data format.
>
> **Table: Comparison with additional bit-plane and VQ methods on LLaMA2-7B** in our response to Reviewer Z3hJ.
>
> ---
>
> > Q1: extended to activation
>
> > A4: Partially, yes. While Bit-Plane Decomposition (BPD) is suitable for decomposing activations, applying the Hessian-based procedure is too expensive for dynamic activations.
>
> ---
>
> > Q2: uint8 as init
>
> > A5: As observed in our Qwen2.5-7B experiments, choosing uint8 as the initialization yields the best WikiText-2 PPL and the highest sum across the six accuracy tasks. Specifically, if the initialization bit-width is too low (e.g., Init4), it causes a loss of original information, leading to the worst performance. Conversely, if the initialization bit-width is too high (e.g., Init16), it overly focuses on preserving the most salient weight representations, but the comprehensive performance still falls short of uint8 (Init8). Thus, uint8 provides the optimal balance.
>
> **Table: Initial Decomposition Bits selected by BPD** in https://anonymous.4open.science/r/START-02D2
>
> ---
>
> > Q3: in table2, VPTQ
>
> > A6: Because Ministral-3 and Qwen-3 are newly released models, VPTQ does not support them. For a broader comparison with VPTQ, please refer to A3.
>
> ---
>
> > Q4: in table3, the latency
>
> > A7: Please refer to BPDQ (SQ) vs. VPTQ (VQ).
>
> Thank you again for your comprehensive review. For your queries, we carefully prepared the answers. If our response effectively addresses your points, we'd be grateful if you consider further improving your score.

---

> > ### Author Rebuttal · Reviewer_rcAB · 2026-04-03
> >
> > Thank you for the rebuttal. It addresses part of my concerns. However, I still find the gains at W4/W3 relatively marginal, and the comparison with VPTQ remains somewhat insufficient. Therefore, I will keep WR.

---

> > > ### Author Response · Authors · 2026-04-05
> > >
> > > Thank you for your response and acknowledgement.
> > >
> > > **1. The "marginal" W4/W3 improvements:**
> > >
> > > As stated in our Abstract: *"While post-training quantization (PTQ) maintains high fidelity at 4-bit, it deteriorates at 2-3 bits"* and Introduction:*"Accordingly, many recent open-source models release low-bit checkpoints. For example, Qwen3 offers an official 4-bit quantized variant (Qwen Team, 2025), suggesting that 4-bit weight-only quantization preserves high fidelity."* Currently, 4-bit PTQ is mature, with numerous 4-bit GPTQ/AWQ models widely deployed. Because the 4-bit representation space is already sufficient, it is theoretically expected and reasonable that improvements at W4/W3 are relatively marginal. Despite this, BPDQ still achieves optimal or near-optimal results in most W4/W3 cases.
> > >
> > > Our core motivation, clearly stated in the abstract, targets the ultra-low-bit limitation: *"Fundamentally, existing methods enforce a shape-invariant quantization grid (e.g., the fixed uniform intervals of UINT2) for each group, severely restricting the feasible set for error minimization."*
> > >
> > > As shown in Table 1, BPDQ-W2-G256 (**BPW 2.19**) significantly outperforms GPTQ-W2-G32 (**BPW 2.56**). For massive models like 72B, the variable grid effectively unlocks the representation potential and significantly boosts the performance of optimization-based PTQ methods.
> > >
> > > | Model | BPW | Wiki2 | GSM8K | MATH500 | ARC-C | BoolQ | HellaS | MMLU |
> > > | :--- | :--- | :--- | :--- | :--- | :--- | :--- | :--- | :--- |
> > > | GPTQ-W2-G32 | 2.56 | 10.01 | 63.46% | 28.40% | 53.16% | 86.21% | 78.60% | 69.59% |
> > > | BPDQ-W2-G256 | 2.19 | **8.94** | **83.85%** | **39.40%** | **60.24%** | **89.72%** | **81.69%** | **75.89%** |
> > >
> > > *(We will release this Qwen2.5-72B BPDQ-W2-G256 checkpoint)*
> > >
> > > **2. Comparison with VPTQ:**
> > >
> > > You mentioned that BPDQ's **~3x** quantization time is "**not cheap**." However, VPTQ requires **~40x** the quantization time. The practical trade-off is that BPDQ achieves highly competitive ultra-low-bit accuracy while maintaining a vastly lower quantization cost and superior hardware adaptability compared to VQ methods.
> > >
> > > **3. Final Statement on Contributions:**
> > >
> > > We want to reiterate that we are the first to identify the "fixed grid" constraint as the fundamental limitation in ultra-low-bit PTQ and solve it backed by rigorous Hessian-based derivation.  As summarized in our contributions:
> > > *   **Insight:** We identify the shape invariance of fixed quantization grids as the fundamental constraint on optimization-based PTQ in low-bit regimes.
> > > *   **Methodology:** We formulate a rigorous optimization framework that extends optimization-based PTQ to variable grids.
> > > *   **Performance:** Extensive experiments across language understanding, reasoning, and long-context tasks demonstrate BPDQ's consistently high fidelity in low-bit regimes. Furthermore, we confirm that BPDQ inherently preserves critical outliers through activation analysis.
> > >
> > > Throughout our experiments, the comparison between GPTQ (fixed grid) and BPDQ (variable grid) serves as the most comprehensive ablation study. Both share the same Hessian framework. Breaking the fixed-grid limitation to rescue the 2-bit accuracy collapse is exactly what we want to present to the community (without mixed-precision, extra inference structures or fine-tuning).
> > >
> > > ---
> > >
> > > **Global Updates During Rebuttal:**
> > >
> > > We respectfully invite you to review the new experiments and insights we provided to other reviewers during this rebuttal phase.
> > >
> > > *   **Comprehensive AnyBCQ Baselines:** As detailed in our response to Reviewer Z3hJ, we evaluated BPDQ on AnyBCQ's models (Llama3.1, Gemma2, and Phi4). BPDQ consistently achieves better 2-bit accuracy. Notably, AnyBCQ takes 9 to 14 times longer to quantize than BPDQ.
> > >
> > > *   **Extension to MoE Architectures:** Reviewer qvHN acknowledged our additional experiments extending BPDQ to MoE architectures. This proves that BPDQ is a generalizable framework capable of handling diverse model structures.

---

### Decision · Program_Chairs · 2026-04-30

**Decision:**

Accept (regular)

**Comment:**

Standard group-scaled M-bit representations convert a vector of g bf16 entries into one bf16 scale s and a vector of g intM (or FP-M) entries. BPDQ proposes a natural extension: represent them as scale1 * int1_tensor1 + scale2 * int2_tensor2 + ... (M terms). It is a reasonable idea, though both I and reviewers have many questions about both the rate overhead and computational overhead (e.g. this method does not allow for using low-precision matmuls, and it's not clear summing multiple bf16 scalars with 1bit multipliers is faster than LUTing a normal VQ). I also recommend authors to study classical quantization literature (shaping, granular errors) to support insights. Comparisons are not very honest since the author's algorithm is "slower inference" compared to GPTQ/AWQ. Comparisons should be to other slow algos like AQLM, PV-tuning, QuIP#/NestQuant/2603.11021 (in general absense of lattice-based algos is a glaring omission).

Despite all these criticisms, I think the idea has merits and should be presented, so I recommend acceptance.